# Lidar detection of high concentrations of ozone and aerosol transported from Northeast Asia over Saga, Japan

Osamu Uchino[1, 2], Tetsu Sakai[2], Toshiharu Izumi[2], Tomohiro Nagai[2], Isamu Morino[1], Akihiro Yamazaki[2], Makoto Deushi[3], Keiya Yumimoto[2], Takashi Maki[2], Taichu Y. Tanaka[2], Taiga Akaho[4], Hiroshi Okumura[4], Kohei Arai[4], Takahiro Nakatsuru[1], Tsuneo Matsunaga[1], Tatsuya Yokota[1]

[1]National Institute for Environmental Studies, 16-2 Onogawa, Tsukuba, Ibaraki 305-8506, Japan

[2]Meteorological Research Institute, 1-1 Nagamine, Tsukuba, Ibaraki 305-0052, Japan

[3]Japan Meteorological Agency, 1-3-4 Otemachi, Chiyoda-ku, Tokyo 100-8122, Japan

[4]Saga University, 1 Honjou, Saga, Saga 840-8502, Japan

*Correspondence to*: O. Uchino (uchino.osamu@nies.go.jp)

**Abstract.** To validate products of the Greenhouse gases Observing SATellite (GOSAT), we observed vertical profiles of aerosols, thin cirrus clouds, and tropospheric ozone with a mobile lidar system that consisted of a two-wavelength (532 and 1064 nm) polarization lidar and a tropospheric ozone Differential Absorption Lidar (DIAL). We used these lidars to make continuous measurements over Saga (33.24°N, 130.29°E) during 20–31 March 2015. High ozone and high aerosol concentrations were observed almost simultaneously in the altitude range 0.5–1.5 km from 03:00 to 20:00 Japan Standard Time on 22 March 2015. The maximum ozone volume mixing ratio was ~110 ppbv. The maxima of the aerosol extinction coefficient and optical depth at 532 nm were 1.2 km$^{-1}$ and 2.1, respectively. Backward trajectory analysis and the simulations by the Model of Aerosol Species IN the Global AtmospheRe (MASINGAR) mk-2 and the Meteorological Research Institute Chemistry-Climate Model, version 2 (MRI-CCM2) indicated that mineral dust particles from the Gobi Desert and an air mass with high ozone and aerosol (mainly sulfate) concentrations that originated from the North China Plain could have been transported over the measurement site within about two days. These high ozone and aerosol concentrations impacted surface air quality substantially in the afternoon of 22 March 2015. After some modifications of its physical and chemical parameters, MRI-CCM2 approximately reproduced the high-ozone volume-mixing ratio. The MASINGAR mk-2 successfully predicted high aerosol concentrations, but the predicted peak aerosol optical thickness was about one-third of the observed value.

## 1 Introduction

Tropospheric ozone is a major air pollutant and impacts human health and vegetation (HTAP, 2010; Yue and Unger, 2014). It is also an important greenhouse gas (IPCC, 2013). Tropospheric aerosols are also air pollutants and aggravate respiratory conditions (HTAP, 2010). Tropospheric aerosols also enhance radiative forcing in a negative (sulfuric acid particles) or positive (black carbon) way (IPCC, 2013), and they affect remote sensing such as the measurement of greenhouse gases from space (Houweling et al., 2005; Uchino et al., 2012a). It is therefore very important to monitor tropospheric ozone and aerosols and to understand their temporal and spatial variations. On the one hand, aerosols transported from East Asia to western Japan have been observed by lidar, and their vertical distributions have been reported (Iwasaka et al., 1988: Murayama et al., 2001; Hara et al., 2009). On the other hand, ozone from Asia has been studied mainly via surface measurements (Akimoto et al., 1996; Yamaji et al., 2006). Continuous ozone vertical distributions measured by DIfferential Absorption Lidar (DIAL) are very useful for studying transport processes and the origin of the ozone.

To validate products of the Greenhouse gases Observing SATellite (GOSAT), we developed a two-wavelength (532 and 1064 nm) polarization lidar (hereafter abbreviated as Mie lidar) to observe vertical profiles of tropospheric and stratospheric aerosols and thin cirrus clouds at the National Institute for Environmental Studies (NIES), Tsukuba (36.05°N, 140.13°E), Japan in 2009. In 2010 we also developed a DIAL to measure tropospheric ozone profiles (hereafter abbreviated as ozone DIAL). The ozone DIAL was installed in a container with the Mie lidar. In March 2011, we moved the lidar container to Saga (33.24°N, 130.29°E) in the Kyushu district of western Japan at a location 2.6 m above sea level. The ozone DIAL was modified in September 2012 (Uchino et al., 2014).

Mie lidar has been used to demonstrate the influence of high-altitude aerosols and cirrus clouds on the GOSAT product of the column-averaged dry air mole fraction of carbon dioxide ($XCO_2$) retrieved from the Thermal And Near infrared Sensor for carbon Observation-Fourier Transform Spectrometer (TANSO-FTS) Short-Wavelength InfraRed (SWIR) spectral data onboard GOSAT. The $XCO_2$ data were improved by taking the vertical profiles of aerosols and cirrus clouds measured by Mie lidar into account (Uchino et al., 2012a). The increases of stratospheric aerosols caused by the 2009 Sarychev eruption and the 2011 Nabro eruption were observed by Mie lidar (Uchino et al., 2012b).

Ozone DIAL has been used to validate the GOSAT ozone product retrieved from TANSO-FTS Thermal InfraRed (TIR) spectral data (Ohyama et al., 2012), to observe ozone concentrations in the lower troposphere, and to compare the observed concentrations with those predicted by the Meteorological Research Institute Chemistry-Climate Model, version 2 (MRI-CCM2) (Deushi and Shibata, 2011). Use of Mie lidar and ozone DIAL will facilitate satellite product validation not only for GOSAT but also for upcoming satellites such as the TROPOspheric Monitoring Instrument (TROPOMI, Veefkind et al., 2012) and the Geostationary Environment Monitoring Spectroscopy (GEMS, Bak et al., 2013). High ozone episodes in the lower troposphere have been observed by lidar (Banta et al., 1998; Koutidis et al., 2002; Ancellet et al., 2005; Eisele and Trickl, 2005; Kuang et al., 2011). Those observational records were

limited to at most one week. We made an 11-day continuous record during 20–31 March 2015.
In this paper we report an event during which high concentrations of ozone and aerosols were observed
almost simultaneously below an altitude of 1.5 km over Saga on 22 March 2015 by Mie lidar and ozone
DIAL. That event substantially impacted surface air quality. We also compared the observational results
with those simulated by the models.


**2 Characteristics of the lidar system and observed parameters**

Mie lidar and ozone DIAL were installed in a container with dimensions of about 228 cm (width), 683 cm
(length), and 255 cm (height), as shown in Fig. 1. Mie lidar is a two-wavelength (532 and 1064 nm)
polarization lidar based on a neodymium-doped yttrium-aluminum-garnet (Nd:YAG) laser; the
characteristics are summarized in Table 1. The output energy at 532 and 1064 nm was 130 mJ, with a
pulse repetition rate of 10 Hz. The diameter of the receiving telescope was 30.5 cm. The output signals
from the photomultiplier tubes (PMT) and a silicon avalanche photodiode (APD) were processed by
transient recorders with a 12-bit analog/digital converter and a photon counter.
The data analysis methods of Mie lidar and ozone DIAL have been described by Uchino et al. (2012b)
and Uchino et al. (2014), respectively. We summarize the observation parameters obtained by Mie lidar.
The backscattering ratio $R$ is defined as

$$R = (BR + BA)/BR, \qquad\qquad\qquad (1)$$

where $BR$ and $BA$ are the Rayleigh and Mie backscattering coefficients, respectively. Backscattering ratio
profiles were derived by the inversion method (Fernald, 1984). The reference altitude was usually set
between 9 and 12 km, where only molecular backscattering could be assumed in the absence of clouds.
We assumed the lidar ratio $LR$ (extinction-to-backscatter ratio) for aerosols to be 50 sr at 532 nm and 45
sr at 1064 nm based on the lidar ratios for Asian dust and pollutant aerosols summarized by Sakai et al.
(2003), Anderson et al. (2003), and Cattrall et al. (2005). The following is a summary of their lidar ratios
at 532 nm:
•   Asian dust $47 \pm 18$ sr (Sakai et al., 2013);
•   dust (spheroids) $42 \pm 4$ sr, Southeast Asia pollution $58 \pm 10$ sr (Cattral et al., 2005);
•   ACE-Asia pollution (fine-dominated, submicron portion) $50 \pm 5$ sr, dust (coarse-dominated, dust-
like chemistry, super-micron portion) $46 \pm 8$ sr (Anderson et al., 2003).
As a simplification, we used the same value for both Asian dust and pollutant aerosols. To calculate $BR$,
we used the atmospheric molecular density profiles obtained by operational radiosondes at the Fukuoka
District Meteorological Observatory (33.58°N, 130.38°E), Japan Meteorological Agency (JMA). The
aerosol extinction coefficient was calculated by multiplying $BA$ by $LR$.
The total volume depolarization ratio $D$ was defined as

$D = S / (P + S) \cdot 100$ (%),        (2)

where $P$ and $S$ are the parallel and perpendicular components of the backscattered signals, respectively.
The particle depolarization ratio $D_p$ was obtained from the equation

$D_p = (D \cdot R - D_m)/(R - 1)$,        (3)

where $D_m$ is the atmospheric molecular depolarization ratio. We used a $D_m$ value of 0.37% for this lidar
system; we calculated $D_m$ from the spectral transmission data of the interference filter at 532 nm and the
Rayleigh backscattering cross sections (Sakai et al., 2003). The value of $D_p$ indicates whether the particles
are spherical or non-spherical; large values indicate the presence of non-spherical particles. The
backscatter-related Ångström exponent $Alp$, the qualitative indicator of aerosol particle size, is defined by

$BA(\lambda) \propto \lambda^{-Alp}$,        (4)

where $\lambda$ is the wavelength. Larger values of $Alp$ indicate the predominance of smaller (i.e.,
submicrometer-sized) particles.  The vertical resolution of these observational parameters was 150 m, and
the time resolution was set to be 1 h for comparison with the Model of Aerosol Species in the Global
Atmosphere (MASINGAR)-mk2 (Yukimoto et al., 2012). The lowest altitude of Mie lidar measurement
was 225 m due to the imperfect overlap of the transmitter-receiver optical axes of the lidar system.

The ozone DIAL consisted of a Nd:YAG laser and a 2-m-long Raman cell filled with $CO_2$ gas that

generated four Stokes lines from stimulated Raman scattering by $CO_2$ ; the characteristics are summarized
in Table 2. In this study, we used three Stokes lines (276, 287, and 299 nm). The output energies of these
Stokes lines were about 8–9 mJ per pulse, with a pulse repetition rate of 10 Hz. The receiving telescope
diameters were 10 cm for boundary layer ozone measurements and 49 cm for free tropospheric ozone
measurements. The Mie lidar and ozone DIAL were synchronized by two pulse-delay generators.

The 276/287 nm and 287/299 nm wavelength pairs were used for ozone DIAL measurements in the

altitude ranges 0.57–2.0 km and 2.0–6.0 km, respectively. The effective vertical resolutions were 270 m
for 0.57–2.0 km and 540 m for 2.0–6.0 km, respectively (Uchino et al., 2014). The time resolution was set
to 1 h to facilitate comparison with the MRI-CCM2. An aerosol correction was not made for ozone
retrieval. Next, we report the continuous lidar observational results made at Saga from 20 March to 31
March 2015.


**3 Ozone DIAL data**

Figure 2a shows a time-altitude cross-section of ozone volume mixing ratios observed by DIAL at Saga
from 11:10 JST on 20 March to 14:33 JST on 31 March 2015. Lidar observations were not obtained from
15:56 JST on 27 March to 21:58 JST on 29 March 2015, mainly because conditions were rainy or cloudy.
We made quality checks of the DIAL data. The gray regions in Fig. 2a correspond to areas where there
were no observational data or the errors were larger than 10%. The errors were computed from the lidar
signal-to-noise ratios by use of Poisson statistics. Regions surrounded by a black rectangle are areas
where the data were affected by aerosols and/or clouds with $R$ larger than 2 at 299 nm. We calculated R
assuming $LR$ = 50 sr, without correcting for attenuation by ozone absorption. In the lowest row of Fig. 2a,
we show hourly data of surface oxidant volume mixing ratios (Ox) at Takagimachi in Saga city measured
by the Saga Prefectural Environmental Research Center (https://www.pref.saga.lg.jp/web/at-
contents/kankyo1/shisetsu/_40810/_41304/_67819.html). Takagimachi is located about 2.8 km northeast
from the ozone DIAL site. Because the surface Ox was observed by a UV photometer, the contribution of
other components such as peroxyacetyle nitrate (PAN) to oxidant concentrations was extremely low, and
the oxidant volume-mixing ratio was considered to be that of ozone.

Figure 2a indicates that the ozone volume mixing ratios measured by DIAL were usually about 50–70

ppbv during the study period. Comparatively high ozone concentrations, >75 ppbv, were detected at
altitudes of 0.57–3 and 0.57–2 km on 20–23 March and 30–31 March, respectively. Notably high ozone
volume mixing ratios of 90–110 ppbv at altitudes of 0.57–1.5 km were observed from 03:00 to 20:00 JST
on 22 March. These high ozone concentrations were also seen in the surface photochemical oxidants data,
i.e., the Ox equaled 92–101 ppbv from 15:00 to 21:00 JST on 22 March, as shown in the lowest row in
Fig. 2a. The maximum concentration of Ox was 101 ppbv at 16:00 JST. This maximum value was far
above the environmental quality standard of 60 ppbv for hourly photochemical oxidants in Japan
(https://www.env.go.jp/en/air/aq/aq.html).

**3.1 Comparison of DIAL data with MRI CCM-2**

The MRI-CCM2 is a global model that simulates chemical and physical processes that affect the
distribution and evolution of ozone and other trace gases from the surface to the stratosphere (Deushi and
Shibata, 2011). Uchino et al. (2014) have provided an outline of MRI-CCM2. The vertical resolution of
the model increases from about 100 to 600 m from the surface to 6 km. The time step of the transport
(chemistry) scheme is 30 (15) min. We used hourly model output data. The horizontal resolution is about
110 km. We examined whether or not the model could simulate DIAL observational results. The MRI-
CCM2 simulated the DIAL observations reasonably well. However, the MRI-CCM2 predicted high ozone
concentrations of 50–60 ppbv and could not reproduce the concentrations of 90–110 ppbv observed with
DIAL below an altitude of 1.5 km during 03:00–20:00 JST on 22 March 2015.

We therefore performed some simulations in which we changed the emission inventory data and the

term that forced the reanalysis wind field. The most reasonable results, shown in Fig. 2b, were obtained
when the following changes were made. The e-folding time of the nudging term was changed from 18
hours to 12 hours to more strongly force the simulated wind fields toward the reanalysis data. In addition,
we changed the emission inventory of Regional Emission inventory in Asia version 1.1 (REAS 1.1)
(Ohara et al., 2007) to the REAS 2.1 emission inventory in 2007 (Kurokawa et al., 2013) and the
$NO_2/NO_x$ emissions ratio from 5% to 15% by volume, which is within the range of uncertainty (Carslaw,
2005). The emission inventory of $NO_x$ increased about 50% from REAS 1.1 to REAS 2.1. Figure 2c
shows the differences between the observed and simulated ozone mixing ratios. Simulated ozone volume
mixing ratios were about 60–70 ppbv below an altitude of 1.5 km from 14:00 JST on 21 March to 21:00
JST on 22 March 2015, lower by about 20–50 ppbv compared with the DIAL results. Moreover, the MRI-
CCM2 predicted high ozone concentrations a half-day earlier than the DIAL observations.

The maximum bias (systematic error) of ozone DIAL data caused by aerosols was estimated to be 20%

(15 ppbv) at 0.57 km; the mean bias and its standard deviation were $7 \pm 5$% in the altitude range 0.57–2.0
km at 11:00 JST.  These biases were estimated from *Alp* observed at the same time by Mie lidar and
assuming $LR = 50$ sr in the wavelength range 276–299 nm based on equations (6) and (7) in Uchino and
Tabata (1991). These biases were not large because the 276/287 nm and 287/299 nm wavelength pairs are
suitable for measurements of ozone in the boundary layer and the free troposphere, respectively
(Nakazato et al., 2007). As mentioned earlier, ozone DIAL data with a statistical error smaller than 10%
were used in this study. Therefore, the uncertainty of the ozone DIAL data was estimated to be smaller
than 22%, and the mean value of the uncertainty was 12%. A model with higher horizontal resolution
might be necessary to more realistically simulate high surface ozone concentration events in the planetary
boundary layer.


**4 Mie lidar data**

Figures 3a, 3b, and 3c show time-altitude cross-sections of the backscattering ratio ($R$), the total volume
depolarization ratio ($D$), and the particle depolarization ratio ($D_p$), respectively, observed by Mie lidar at
Saga from 09:24 JST on 20 March to 14:34 JST on 31 March 2015. Mie lidar data were not obtained from
15:56 JST on 27 March to 21:58 JST on 29 March 2015, mainly because conditions were rainy or cloudy.
We made quality checks of Mie lidar data. Gray regions are areas where there were no observational data
or the data were affected by clouds.

Aerosol layers with $R$ in the range 2–4 almost always existed below an altitude of 2.5 km during 20–

31 March 2015. An event of high aerosol loading with large values of $R$ (>8) was observed below
altitudes of 1.5 km during 03:00–21:00 JST on 22 March, when the values of $D$ were small (the mean $\pm$
standard deviation was $3.9 \pm 2.1$%) compared with those before and after the event. The values of $D$ were
larger than $7.9 \pm 2.1$% from 15:00 JST on 21 March through 15:00 JST on 23 March, except for 03:00–
21:00 JST on 22 March. The main aerosol component during the event might have been submicrometer-
sized spherical particles, because $D_p$ was small ($4 \pm 2$%), and the wavelength exponent *Alp* was large (1.3
$\pm 0.3$). In contrast, the main aerosol particles before and after the event may have been supermicrometer-
sized, nonspherical mineral dust particles because $D_p$ was comparatively large ($13 \pm 3$%) and *Alp* was 1.0
$\pm$ 0.2 (Sakai et al., 2003; Cattrall et al., 2005). When there were no clouds above, $R$ at 1064 nm was
estimated assuming $Alp = 1.5$ at the reference altitude, where very small amounts of aerosols were
expected to be present, i.e., $R = 1.06 \pm 0.06$ ($D = 1.2 \pm 0.5$) at 532 nm in the altitude range 3–6 km. If the
value of $Alp$ was changed from 1.0 to 2.0 at the reference altitude, the uncertainty in $Alp$ was estimated to
be $\pm$ 0.2. $Alp$ was 0.3–2.0 in the 11-day Mie lidar record. The maximum errors of $D$ and $D_p$ were 0.1%
and 2% for $R{>}2$ at 532 nm.
During the same time period, high aerosol concentrations were also observed at the surface (Fig. 4).
Hourly values of the mass concentrations of particulate matter with a diameter of 2.5 μm or less (PM$_{2.5}$) at
Takagimachi measured by the Saga Prefectural Environmental Research Center were 23 μg m$^{-3}$ at 10:00
JST and increased up to a maximum value of 110 μg m$^{-3}$ at 15:00 JST on 22 March; the concentrations
were greater than 82 μg m$^{-3}$ during 13:00–16:00 JST and decreased to 17 μg m$^{-3}$ at 01:00 JST on 23
March. The daily mean value of PM$_{2.5}$ was 50.6 μg m$^{-3}$ for 24 hours on 22 March at Takagimachi, larger
than the environmental quality standard of 35 μg m$^{-3}$ in Japan (https://www.env.go.jp/en/air/aq/aq.html).

**4.1 Comparison of Mie lidar data with MASINGAR mk-2**

The MASINGAR-mk2 is an improved version of the MASINGAR aerosol model (Tanaka et al., 2003); it
treats five aerosol species: sulfate, black and organic carbon, sea salt, and soil dust. We used emission
data for sulfur dioxide and for black and organic carbon from MACCity (Granier et al., 2011). Soil dust
and sea salt were represented by 10 bins with particle diameters of 0.2–20 μm. The model was coupled
online with the atmospheric general circulation model MRI-AGCM3 (Yukimoto et al., 2012). The
Meteorological fields were taken from JMA Global Analysis data (GANAL). The horizontal resolution of
the MASINGAR-mk2 was about 60 km, and the number of vertical layers was 40 from the surface to 0.1
hPa. The vertical resolutions were 100, 300, and 600 m at the lowest level and altitudes of 1 and 6 km,
respectively. The time step of the transport (chemistry) scheme was 450 seconds, and we used hourly
model output data.
Figures 5a and 5b show the time-height cross sections of aerosol extinction coefficients observed by
Mie lidar and simulated by MASINGAR-mk2, respectively. Figure 5c represents the difference between
the observed and simulated extinction coefficients. The model was able to capture the general
characteristics of the observational results rather well. A close look at Fig. 5c reveals that the model
underestimated the aerosol extinction coefficients of the anthropogenic pollutant event on 22 March but
slightly overestimated the extinction coefficients associated with particles having larger total volume
depolarization ratios on 30 and 31 March (i.e., dust-dominant case).

**4.2 Comparison of aerosol optical depths**

Figure 6 shows temporal variations of the aerosol optical depths (AOD) measured by Mie lidar at 532 nm
and sky radiometer at 500 nm (Kobayashi et al., 2006, Uchino et al., 2012a) and simulated at 550 nm by
MASINGAR-mk2 from 20 to 31 March. To estimate AODs from the lidar data, the extinction coefficient
at 225 m was extrapolated to the ground, the extinction coefficient from 15 to 35 km was observed at
night on the same day, and $S$ was assumed to be 50 sr for all altitudes. When clouds and thick aerosols
were present, AODs were not obtained. The sky radiometer was positioned on the roof of the building,
which is four stories high and located to the west of the container (brown building in Fig. 1). Although it
must be noted that the measured and simulated wavelengths differed slightly, the AODs were almost the
same, except for the high aerosol and ozone event on 22 March. The mean bias ± the standard deviation
of the AOD between Mie lidar and sky radiometer was $0.029 \pm 0.051$, and that between MASINGAR mk-
2 and sky radiometer was $-0.07 \pm 0.24$ for 20–31 March, except for 12:00–14:00 JST on 22 March. The
maximum values of the AODs were 2.1 at 12:00 JST by lidar, 1.92 at 13:00 JST by sky radiometer, and
0.53 at 13:00 JST by MASINGAR-mk2. One possible reason for the large difference in AOD (~0.2)
between Mie lidar and Sky radiometer data is that we set the reference altitudes to 8.2 km and 2.8 km at
12:00 and 13:00 JST on 22 March, respectively, for the lidar because the backscattered signals were
strongly attenuated by the dense aerosol layers below 2 km. This might have caused errors in the AODs
for the Mie lidar data. The possibility that the views obtained by the sky radiometer and Mie lidar differed
might also account for the difference.
The model underestimated the AODs by factors of about 3.6–4 compared to the sky radiometer and
lidar observations. One plausible reason for that is that the model resolution (about 60 km) was
insufficient to reproduce the observed prominent peak in which the observed AOD increased from 1.0 to
2.0 in 6 hours. The other plausible reason for the underestimation is the uncertainty of the emissions
inventories of aerosol precursors. Grainer et al. (2011) collected various emission inventories and
compared them on a global scale. They found that differences in Chinese sulfur dioxide ($SO_2$) emissions
in 2000 reached 66% between the lowest and highest emissions and concluded that there was no
consensus among the different inventories for the emissions of Chinese $SO_2$. This large variation among
the inventories indicates that there is a large error associated with the estimate of $SO_2$ emission in China.
In their comparison, the MACCity emission, which was used in the MASINGAR-mk2 simulation,
showed the lowest amount of Chinese $SO_2$ emission among the inventories. This might explain the
underestimation of pollutant aerosol (sulfate) concentrations. In the MASINGAR-mk2 simulation, the
dust emission flux was estimated by a parameterized dust emission scheme and was strongly dependent
on various parameters (e.g., soil texture, soil wetness, land use, snow cover fraction, vegetation cover, and
surface wind speed). The dust model intercomparison project (DMIP; Uno et al., 2006) reported that
simulated amounts of dust emissions over East Asia differed sometimes by a factor of ten among eight
dust models (including the former version of MASINGAR). These facts indicate that estimates of dust
emissions are associated with large errors. To solve this problem, for example, it might be better to use
the near real-time satellite data of $SO_2$ and nitrogen dioxide ($NO_2$) provided by the Ozone Monitoring
Instrument (OMI) onboard NASA's Aura satellite (Krotkov et al., 2016), and/or to use a data assimilation
technique that integrates model simulation and observational data (Yumimoto et al., 2016).


**5 Discussion: origin and transport pathways of ozone and aerosol plumes**


Figure 7 shows the time-altitude cross sections of total aerosol extinction coefficients at 550 nm, and the ratios of dust extinction coefficients to total aerosol extinction coefficients simulated by MASINGAR-mk2 with potential temperatures over Saga for 20–31 March 2015. For the event on 22 March, the model predicted dust particles (about 60–100%) in the altitude range 1–3 km, and it predicted sulfate (about 40–60%) and dust (about 30–40%) particles below 1 km. The numbers in parentheses indicate the ratio of each component's extinction coefficient to the total extinction coefficient. The dust particles descended to the surface in the afternoon (Fig.7b). For the event on 30 March, MASINGAR mk-2 predicted dust particles (about 50–100%) at altitudes of 1–6 km, and it predicted sulfate (about 50–80%) and dust (about 0–20%) particles below 1 km in the morning. Mie lidar data supported the model prediction because $D_p$ was high (17 ± 6%) at altitudes of 1–3 km and low (10 ± 3%) below 1 km. For both events, small amounts of organic carbon, black carbon, and sea salt particles were predicted.

To identify the origin of the aerosols and related transport processes, three-dimensional backward trajectories of air parcels were calculated with the NOAA Hybrid Single Particle Lagrangian Integrated Trajectory (HYSPRIT) model (Draxler and Hess, 1998; Stein et al., 2015). Air parcels were initially left at altitudes of 1500 m (Fig. 8a) and 500 m (Fig. 9a) over the lidar site at Saga. The trajectories were calculated for three days from 21:00 UTC on 21 (06:00 JST on 22) March 2015. Figures 8b and 9b show the time-altitude cross sections of dust and sulfate extinction coefficients simulated by MASINGAR mk-2 along the trajectory paths of Figs. 8a and 9a, respectively. Based on the results of the backward trajectories and the model simulations, the dust and sulfate particles on 22 March could have been transported within about two days from the Gobi Desert and the North China Plain (NCP), respectively, to the measurement site. The MASINGAR mk-2 simulation suggested that the dust particles emitted during 18:00–24:00 UTC on 19 March around 40°N and 105°E were responsible for the dust storm captured by the Mie lidar observation. The highest concentrations of $SO_2$ and $NO_2$ in the world were observed in the NCP during 2013–2015 by the Ozone Monitoring Instrument (OMI) onboard NASA's Aura satellite, as shown in Fig. 5 by Krotkov et al. (2016). These gases are important precursors of sulfate particles and ozone. Figure 10 shows horizontal maps of ozone volume mixing ratios at 925 hPa (an altitude of about 760 m) simulated by the MRI-CCM2 at 21:00 JST on 19, 20, and 21 March and at 03:00 JST on 22 March 2015. These maps indicate that the high ozone concentrations could have been transported from the NCP to the Yellow Sea and then to Saga within about two days.

Because it was difficult to obtain observational data of surface ozone and sulfate particles over the NCP, including Beijing, on 19–20 March, we referred to the following papers with respect to those data. According to the ozonesonde measurements made by Wang et al. (2012), ozone concentrations ≥90 ppbv were observed over Beijing, China in late March. Ma et al. (2016) reported a significant increase of surface ozone from 2003 to 2015 at Shangdianzi (40.65°N, 117.10°E), which is located about 100 km northeast of suburban Beijing, and the maximum daily average 8-h concentrations of ozone appear to

have been >100 ppbv in March 2015 based on Fig. 2 in their paper. High $PM_{2.5}$ and submicron aerosol
concentrations have been observed in Beijing (Zhang et al., 2013; Sun et al., 2015). Ozone and aerosol
concentrations may therefore have been high in March 2015 over the NCP.
To elucidate the vertical transport processes of the aerosol and ozone in the lower troposphere over the
measurement site, we show in Fig. 11 the time variations of the heights of the mixed layers from two
hours after sunrise to two hours before sunset during 09:24 JST on 20 March through 14:34 JST on 31
March 2015. These altitudes were estimated from (1) the 1064 nm range-corrected backscatter signals,
with a range resolution of 15 m and a time interval of 20 min, using the wavelet covariance transform
method (Baars et al., 2008; Izumi et al., 2016) and (2) signals obtained from radiosonde data at Fukuoka
and the JMA Meso-scale Analysis (MA) data over Saga using the parcel method (Holzworth, 1964).
When the mixed layers developed in the afternoon, the heights of the mixed layers (1.5–2 km) estimated
by Mie lidar were similar to those estimated by MA. Although the radiosonde data at 9:00 JST on 22
March found the height of the mixed layer to be 117 m (Stull, 1988), it was difficult for Mie lidar to
detect the mixed layer because the lowest altitude of the Mie lidar measurement was 225 m.
The dust particles that originated from the Gobi Desert arrived at altitudes of 1–3 km over the lidar site
at 06:00 JST on 22 March. When the mixed layer developed to 1.5–2 km at 11:00–15:00 JST on 22, the
dust particles were assumed to be mixed into the boundary layer and then to reach the surface by
entrainment, as simulated in Fig. 7b. This may explain the sharp increase in PM2.5 concentrations at the
surface after 11:00 JST, as shown in Fig. 4. A similar phenomenon was observed over the northern
Kyushu area during the dust event in late May–early June 2014 (Uno et al., 2016). A similar high-surface-
ozone event was observed by eight ozonesonde measurements during 6–9 June 2003 over the Seoul
metropolitan region (Oh et al., 2010).


**6. Concluding remarks**

By using ozone DIAL and a two-wavelength polarization (Mie) lidar, we made continuous measurements
of ozone and aerosol concentrations over Saga during 20–31 March 2015. High ozone and high aerosol
concentrations that occurred nearly simultaneously were observed in the altitude range 0.5–1.5 km from
03:00 to 20:00 JST on 22 March 2015. The ozone volume mixing ratio was larger than 100 ppbv. The
aerosol extinction coefficient and AOD at 532 nm were larger than 0.5 $km^{-1}$ and 1.5, respectively.
Backward trajectory analysis and the simulations by the MASINGAR mk-2 and the MRI-CCM2
models indicated that mineral dust particles from the Gobi Desert and an air mass with high ozone and
aerosol (mainly sulfate) concentrations that originated from the North China Plain could have been
transported over the lidar site within about two days. Based on the lidar and surface measurement data
and the simulation by the MASINGAR-mk2, there is a possibility that the air mass with high ozone and
aerosol concentrations could have been transported from the lower troposphere to the surface by vertical
mixing when the planetary boundary layer developed in the afternoon of 22 March 2015. The
combination of ozone DIAL measurements with surface in-situ ozone measurements is very useful for
studying the process of descent of high ozone concentrations in the lower troposphere to the surface and
the impacts on surface air quality. Such measurements of pollution plumes that descend from the free
troposphere to the surface are highly recommended (HTAP, 2010).
The MRI-CCM2 could approximately reproduce the high-ozone volume-mixing ratios after some
modifications of physical and chemical parameters. MASINGAR mk-2 successfully predicted high
aerosol concentration events, but the predicted peak AOD was about one-third of the observed AOD. For
further improvement of these models, it will be important to continue comparing these models with ozone
DIAL, Mie lidar, and surface in-situ ozone and particle measurements.

*Acknowledgements*.  We used radiosonde data measured by the Japan Meteorological Agency and hourly
concentrations of surface oxidant and $PM_{2.5}$ measured by the Saga Prefectural Environmental Research
Center. The NOAA Hybrid Single Particle Lagrangian Integrated Trajectory (HYSPRIT) model was used
to calculate backward trajectories of air parcels. The authors thank the anonymous referees and editors for
helpful comments and suggestions.

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

**Table 1.** Characteristics of Mie lidar
_______________________________________________________________________________
Transmitter
Laser                              Nd:YAG
Wavelength              532 nm                      1064 nm
Pulse energy            130 mJ                      130 mJ
Pulse repetition rate              10 Hz
Pulse width                        8 ns
Beam divergence         0.2 mrad                    0.2 mrad
_______________________________________________________________________________
Receiver
Telescope type               Schmidt Cassegrain
Telescope diameter              30.5 cm
Focal length                    3048 mm
Field of view                   1 mrad
Polarization            P and S                     None
Number of channels       3                          1
Interference filter
Center wavelength     532.0 nm                    1064.1 nm
Bandwidth (FWHM)      0.29 nm                     0.38 nm
Transmission          0.66                        0 .58
Detectors                  PMT                        APD

(Hamamatsu R3234-01)      (EG&G C30956EH)

Signal processing       12bit A/D + Photon counting
Time resolution         1 min
Vertical resolution     7.5 m
_______________________________________________________________________________











**Table 2.** Characteristics of tropospheric ozone DIAL system

_______________________________________________________________________________

| Transmitter | | | | | |
|---|---|---|---|---|---|
| Pump laser | Nd:YAG | | | | |
| Wavelength | 266 nm | | | | |
| Pulse energy | 107 mJ | | | | |
| Pulse repetition rate | 10 Hz | | | | |
| Pulse width | 8 ns | | | | |
| Raman active gas | $CO_2$ | | | | |
| Stokes lines | 276 nm | 287 nm | 299 nm | 312 nm | |
| Pulse energy | 7.5 mJ | 9.1 mJ | 8.4 mJ | No. meas. | |
| Beam divergence | 0.1 mrad | | | | |

_______________________________________________________________________________

| Receiver | | | | | |
|---|---|---|---|---|---|
| Telescope type | Newtonian | | | Prime focus (fiber coupled) | |
| Telescope diameter | 49 cm | | | 10 cm | |
| Focal length | 1750 mm | | | 320 mm | |
| Field of view | 1 mrad | | | 3 mrad | |
| Interference filter | | | | | |
|   Center wavelength | 287.2 nm | 299.0 nm | 312.0 nm | 276.1 nm | 287.2 nm |
|   Bandwidth (FWHM) | 1.02 nm | 1.15 nm | 0.82 nm | 1.07 nm | 1.05 nm |
|   Transmission | 0.18 | 0.32 | 0.36 | 0.17 | 0.21 |
| Detectors | PMT (Hamamatsu R3235-01) | | | | |
| Signal processing | 12bit A/D + Photon counting | | | | |
| Time resolution | 1 min | | | | |
| Vertical resolution | 7.5 m | | | | |

_______________________________________________________________________________


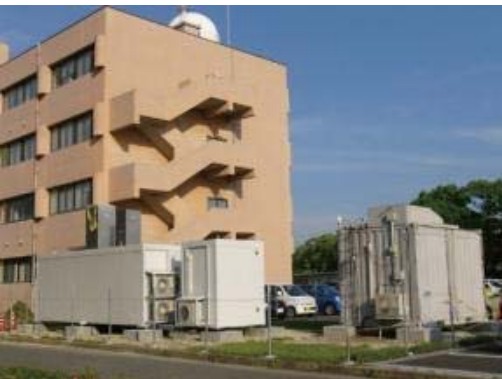 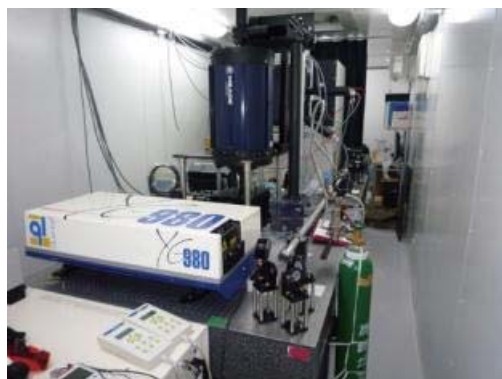

**Figure 1.** Mie lidar and ozone DIAL (right) were installed in the container at the left on the ground (left).

(a)

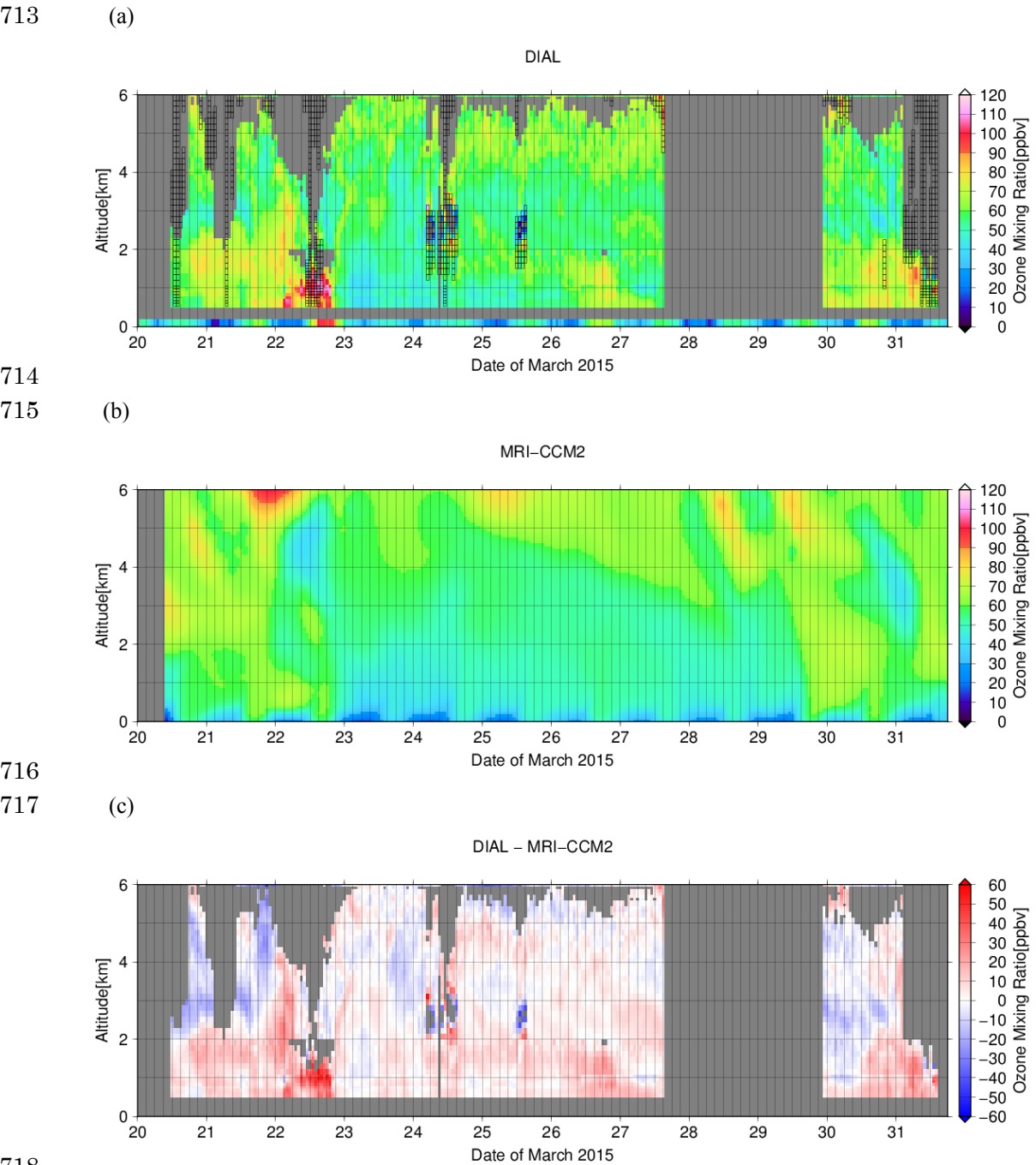


(b)

(c)

**Figure 2.** Time-altitude cross-sections of (a) ozone volume mixing ratios observed by DIAL over Saga
from 11:10 JST on 20 March to 14:33 JST on 31 March 2015, (b) the ratios simulated by a modified
MRI-CCM2 for 20–31 March 2015, and (c) the difference between the observed and simulated ozone
volume mixing ratios (a–b). Gray regions indicate areas where there were no observational data or the
statistical errors were larger than 10%. Regions enclosed with black rectangles are areas where the data
were affected by aerosols and/or clouds. The lowest row in Fig. 2a shows photochemical oxidant (ozone)
volume mixing ratios at Takagimachi in Saga city as measured by the Saga Prefectural Environmental
Research Center.
(a)

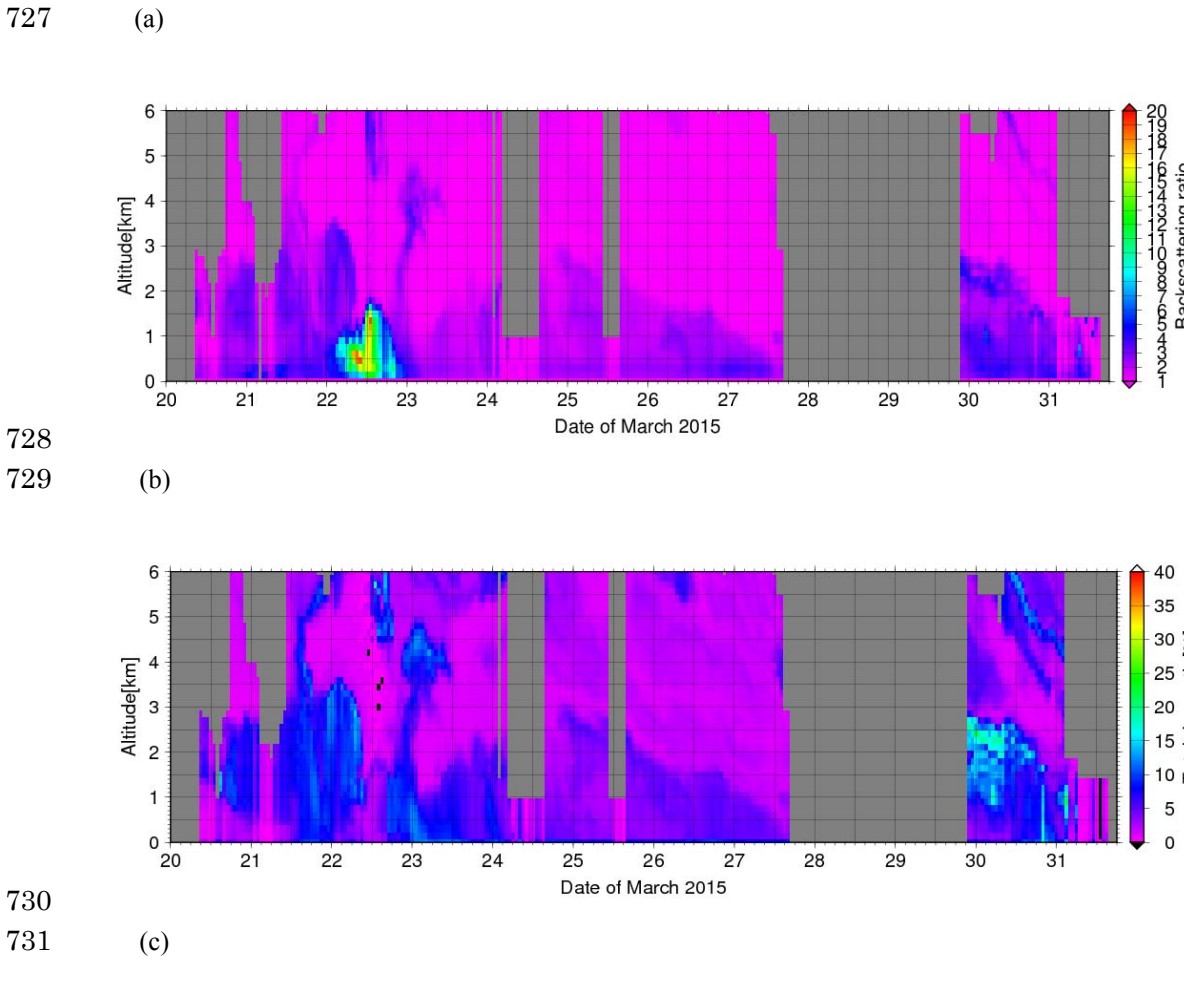


(b)

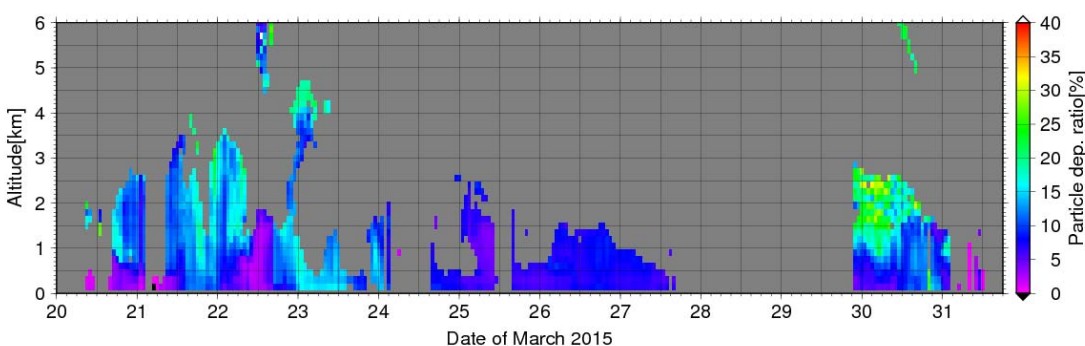


(c)


**Figure 3.** Time-altitude cross-sections of (a) backscattering ratios, (b) total volume depolarization ratios
and, (c) particle depolarization ratios for $R \geq 2.0$ at 532 nm observed by Mie lidar at Saga from 09:24 JST
on 20 March to 14:34 JST on 31 March 2015. Lidar observations were not available from 15:56 JST on
27 March to 21:58 JST on 29 March 2015 mainly because of rainy or cloudy conditions. Gray regions are
areas where there were no observational data or where the observations were affected by clouds.



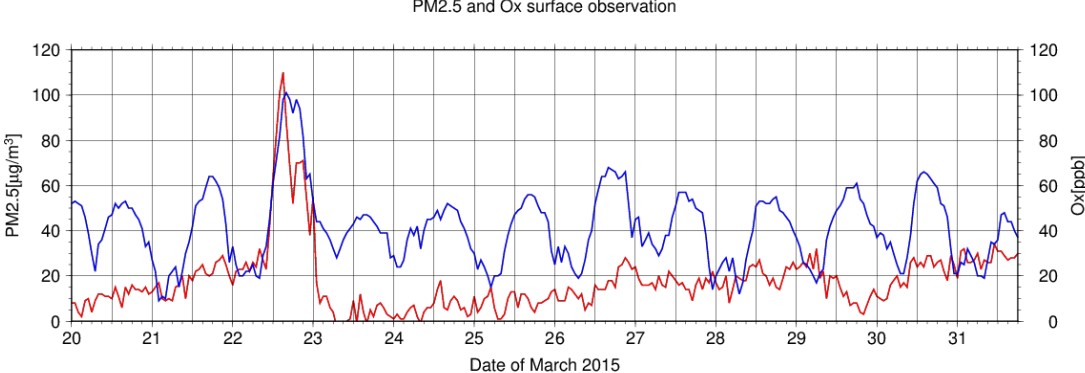


**Figure 4.** Hourly (JST) data of surface PM2.5 (red line) and Ox (blue line) measured by the Saga
Prefectural Environmental Research Center for 20–31 March 2015. The volume mixing ratio of Ox was
considered to be that of ozone.


























(a)

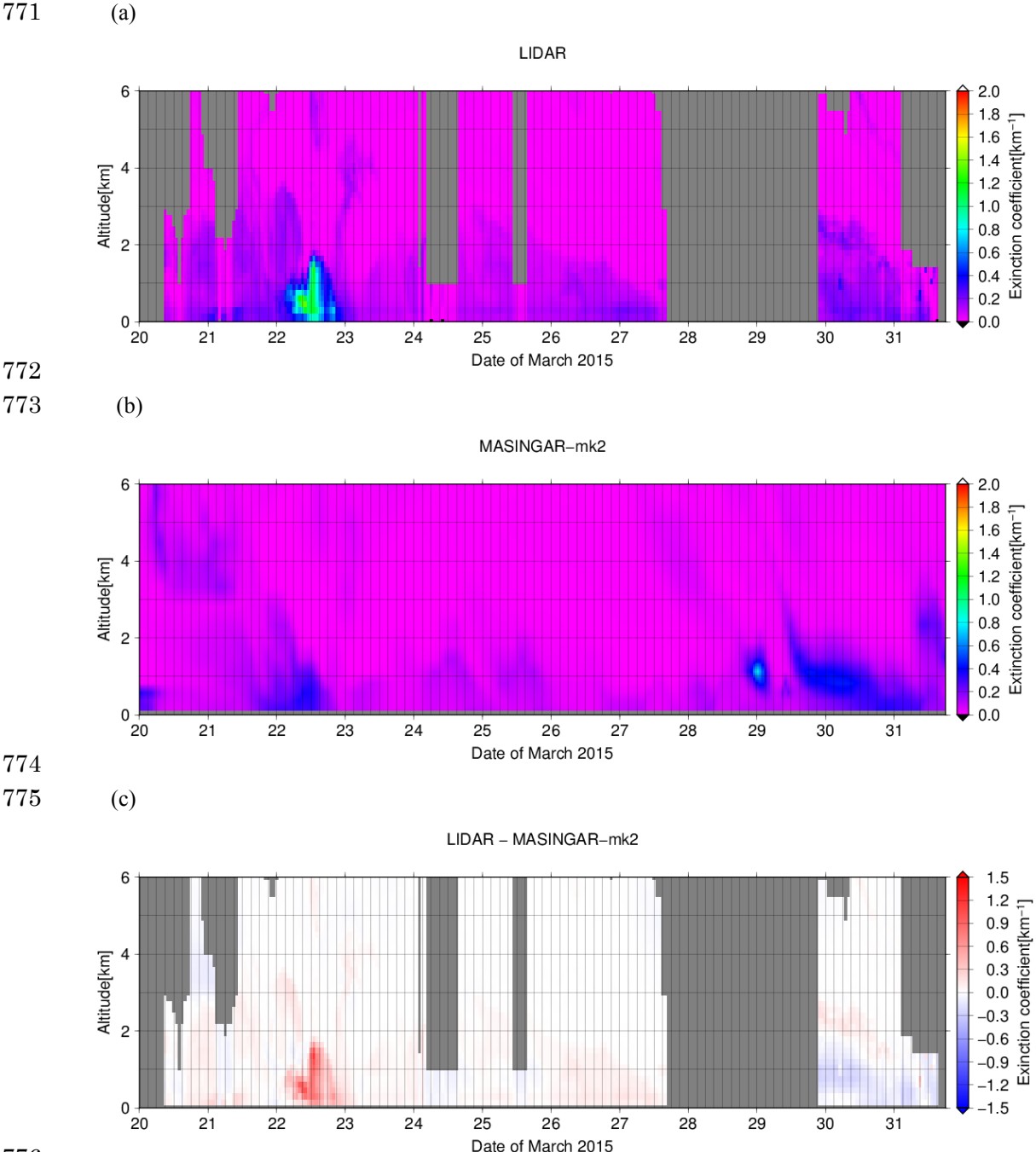

(b)
(c)

**Figure 5**. Time-altitude cross sections of (a) aerosol extinction coefficients observed by Mie lidar at 532 nm over Saga from 09:24 JST on 20 March to 14:34 JST on 31 March 2015, (b) the coefficients simulated by MASINGAR-mk2 at 550 nm for 20–31 March 2015, and (c) the difference between the Mie lidar observations and the simulation (a–b). Gray regions represent areas where there were no observational data.



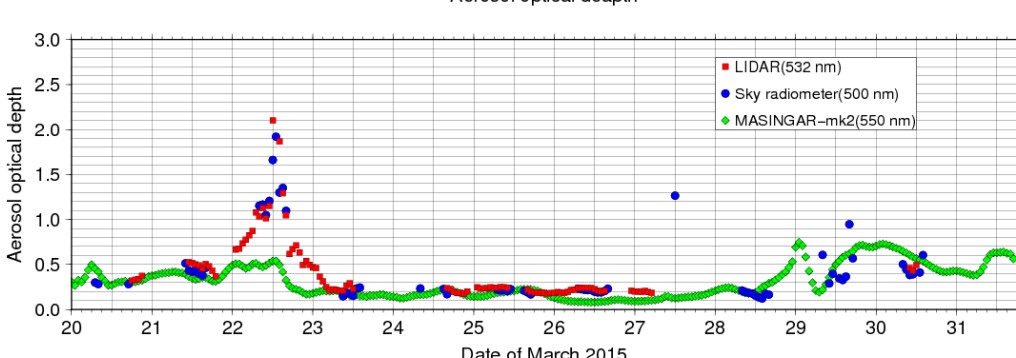


**Figure 6.** Temporal variation of the aerosol optical depth (AOD) measured by Mie lidar at 532 nm (red circles), by sky radiometer at 500 nm (blues circles), and simulated at 550 nm by MASINGAR-mk2 (green circles).



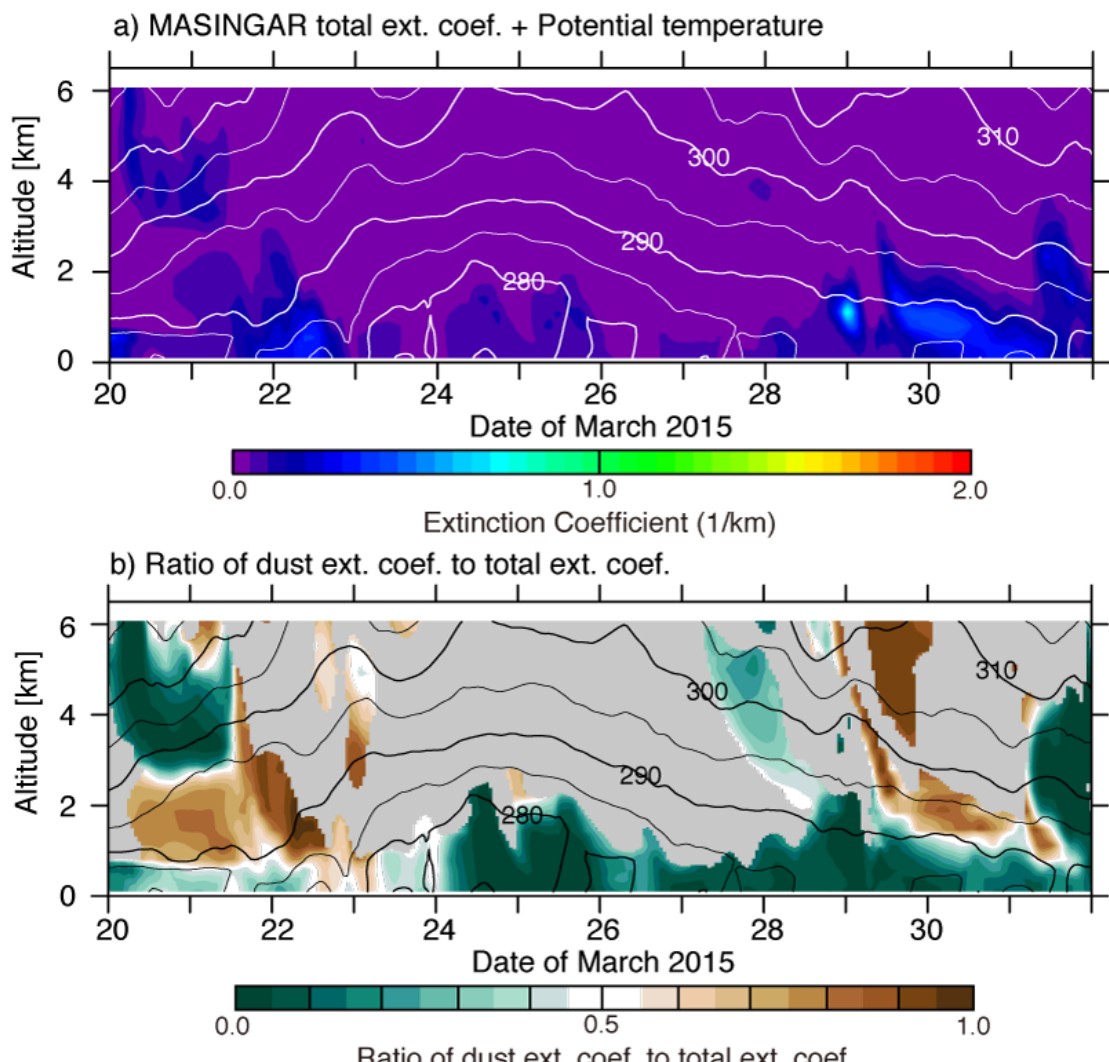



**Figure 7.** Time (JST)-altitude cross sections of (a) total aerosol extinction coefficients at 550 nm (color
shading) and (b) ratios of dust extinction coefficient to total aerosol extinction coefficient (color shading)
simulated by the MASINGAR-mk2 with potential temperatures (black contours) over Saga for 20–31
March 2015. The gray regions in Fig. 7b indicate that the simulated total aerosol extinction coefficient
was less than 0.02.







(a)

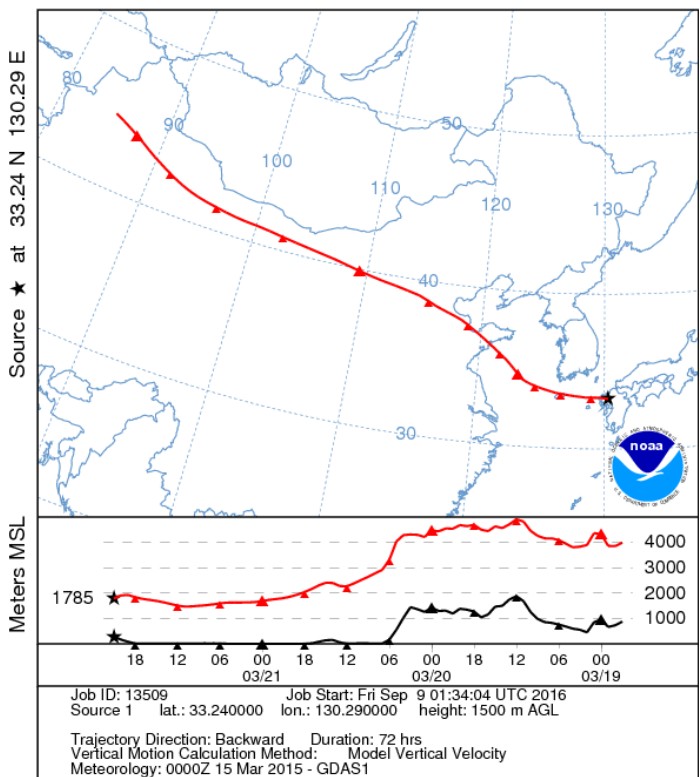


(b)

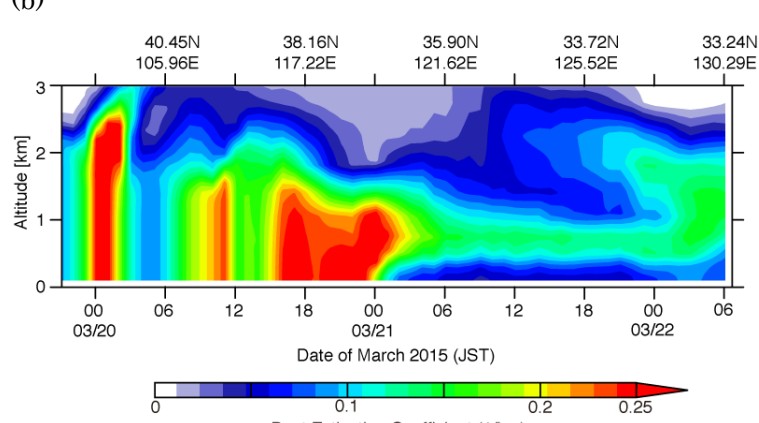


**Figure 8.** (a) The 72-h HYSPLIT-model backward trajectory (red line) and terrain height (black line)
from Saga at 1500 m above ground level (AGL) ending at 06:00 JST on 22 May 2015. (b) Time-altitude
cross section of dust extinction coefficient simulated by the MASINGAR mk-2 along the trajectory path.


(a)

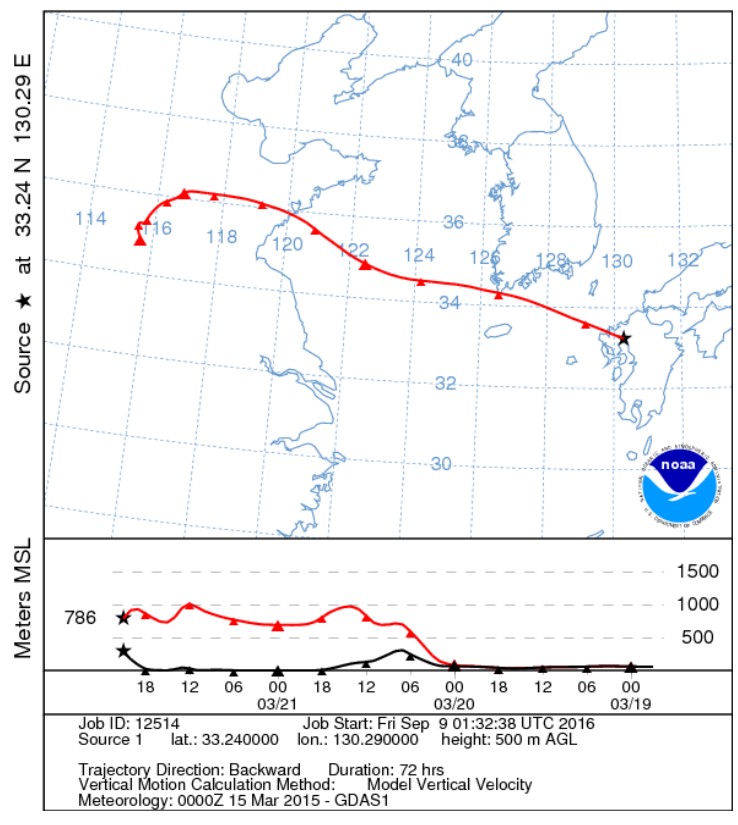



(b)

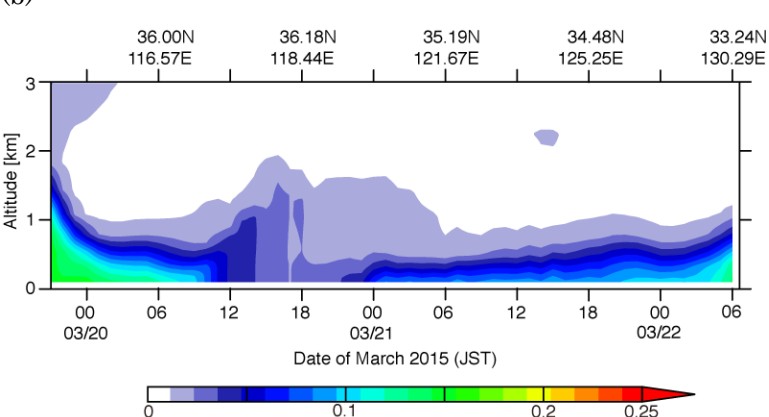



**Figure 9.** Same as Fig. 8, but for AGL at 500 m.


(a)                                    (b)

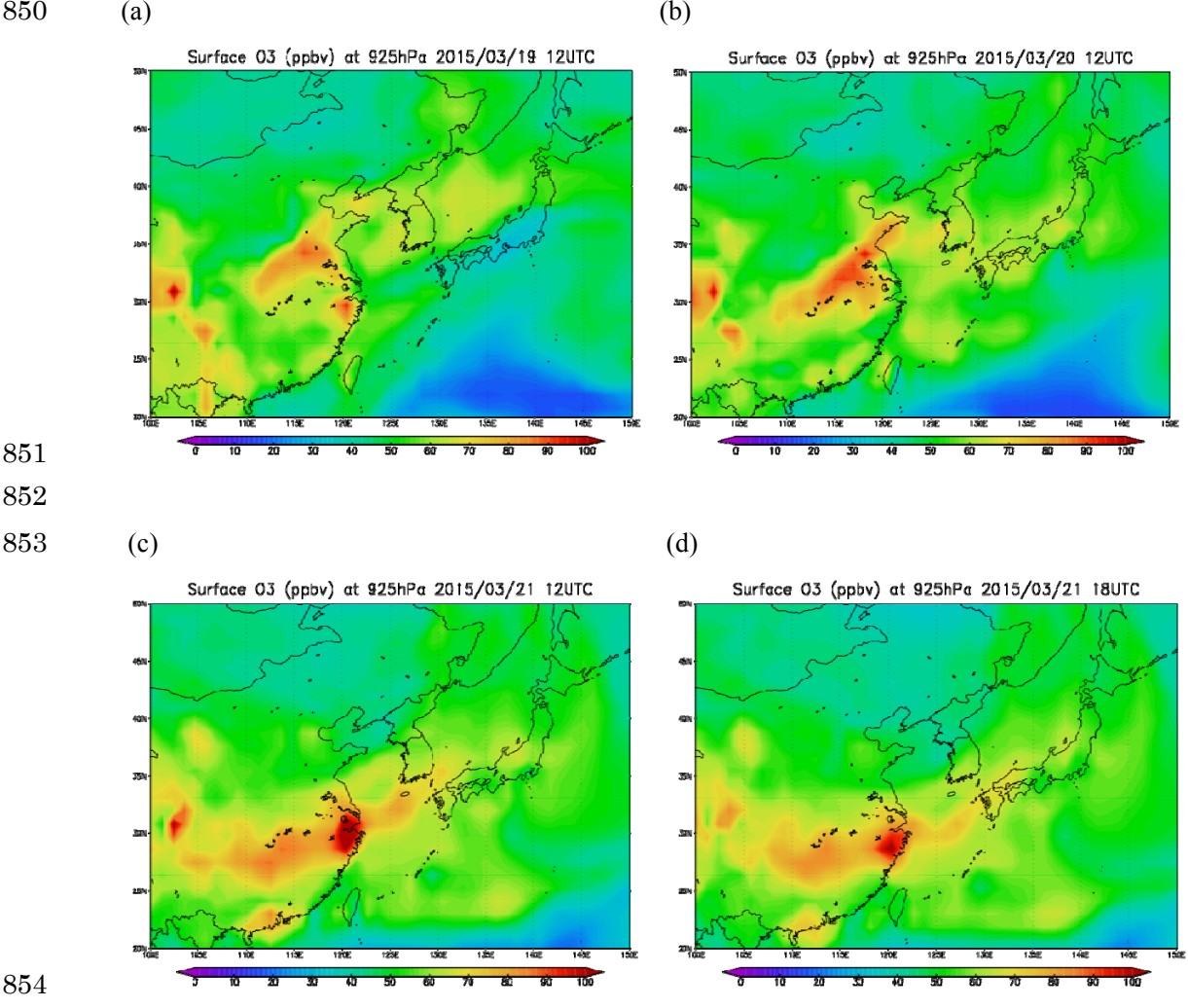

(c)                                    (d)

**Figure 10.** Horizontal maps of ozone volume mixing ratios in ppbv predicted by the MRI-CCM2 for the
925 hPa pressure level (an altitude of about 760 m) at 21:00 JST (JST = UT + 9) on (a) 19, (b) 20, (c) 21
March, and (d) 03:00 JST on 22 March 2015.

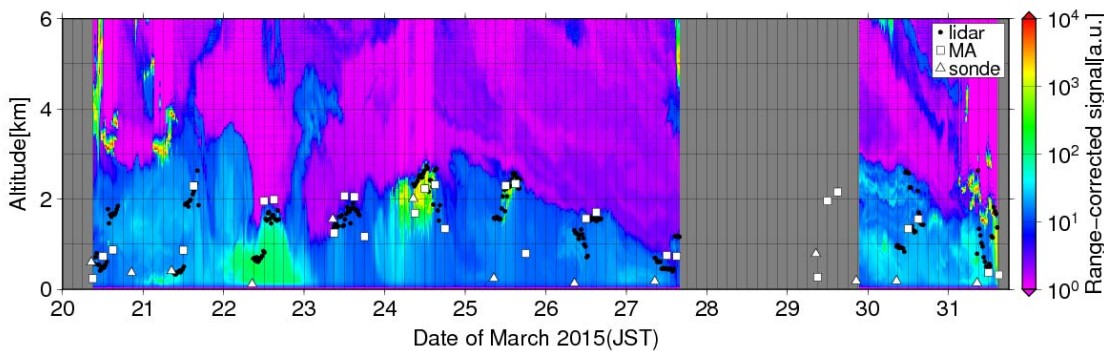

**Figure 11.** Time-altitude cross section of range-corrected backscatter signal at 1064 nm (color shading)
and the heights of the mixed layers estimated by Mie lidar (closed black circles), radiosonde (open
triangles), and JMA Meso-Scale Meteorological Analysis (open squares) data over Saga from 09:24 JST
on 20 March to 14:34 JST on 31 March 2015.