# Peer review of "Lidar detection of high concentrations of ozone and aerosol transported from Northeast Asia over Saga, Japan"

_Atmospheric Chemistry and Physics, 2016_

## Referee Comment (RC1) · Anonymous Referee #1 · 26 Jul 2016

The paper by Uchino et al. presents lidar measurements of high ozone and aerosol concentrations over southern Japan and a comparison with models. Measurements and the model outputs are interesting and a complete analysis of these datasets is worth scientific publication. However, the current analysis of these datasets does not seem to fully exploit the information they provide and many statements are not clearly justified (even though the observational and modelling datasets to do it are potentially available already). I recommend the following major revisions:

1) Evidence on the origins (location and type of source) and transport pathways (particularly in the vertical) of the ozone and aerosol plumes are not clearly provided. Chemistry-transport model outputs are not exploited for this, as currently, simulations

are only used for comparison with lidars. I recommend fully developing this aspect, with analysis of the model outputs at the scale of East Asia and in the vertical. This is very important to provide information on the horizontal and vertical pathways of the pollutant plumes and their origin (e.g. type of aerosols).

2) Analysis of lidar data: the vertical structure of ozone and aerosol layers are not completely analysed, as the heights of the atmospheric boundary layer (mixing and residual layers) are not depicted. As shown in numerous papers, the detection of these layers may be easily done with the aerosol lidar. Such analysis should be added in the paper and precisely explain in a chronological order the mechanisms involved (i.e. vertical mixing, arrival of pollutants, etc).

3) Technical characteristic of the datasets: in the presentation of the results, too much technical information of the datasets (instrument characteristics, model configuration, variables describing the datasets, etc) is given. The geophysical interpretation of the measurements is relatively scarce. For sake of clearness for the readers, I recommend that most of technical remarks on the datasets are given a previous section dedicated to datasets characteristics and then only the geophysical interpretation is given when describing the figures.

4) The extinction-to-backscatter ratio : the climatological values that are used are expected to be suited to a particular type of aerosol (it changes a lot depending on the origin and size). What kind of aerosol is it for 50 sr at 532 nm? Measurements suggest the presence of large non-spherical particles (likely desert dust). How is this taken into account? I recommend using different extinction-to-backscatter ratios for each kind of particles (dust, sulphate, etc).

I recommend the following minor revisions:

1) English language should be revised in the whole the paper.

2) Colour scales in figure 2 should be changed in order to highlight changes from

background.

3) Line 93: what is the meaning of "The errors were equated to 100% times the lidar signal-to-noise ratios". Please clarify.

4) Lines 106-108 : The justification of the following statement is not clear : "These observational results show the descent of regions of high ozone concentrations and suggest that air with high ozone concentrations was transported to the surface by vertical mixing when the planetary boundary layer developed during the daytime". What is the height of the boundary layer? This height should be clearly shown in the figures. When was it mixed? Which elements suggest a "descent" or downward mixing of the high ozone plume? This should be thoroughly explained and justify. The model may be used to describe this mixing mechanism. When it is mentioned "daytime" vertical mixing, it is in which day? In which location vertical occurs? The current explanation is very scarce and unclear. This should be thoroughly explained and justify.

5) Line 120: The sentence "The most reasonable results were obtained when the following changes were made (Fig. 2b)" does not seem to match the Figure as it does not explain the changes in the model, but it presents the final results.

6) Line 130: there is also a difference in the time of the arrival of the plume. Please, report.

7) The term "bad weather" is not objective. I suggest to use "rainy" conditions or similar.

8) The wavelength exponent, Alp : the name to this variable does not seem very conventional. For optical depth, the term "Angström exponent" is usually used in literature. The name Âń wavelength exponent Âż does not seem very explicit. Why the symbol is "Alp" ? Is this term often used by the scientific community?

9) Line 167 : The term "always existed" is not clear. Does it mean that it is observed in the timeseries shown in Figure 3 ? Please, clarify.

10) What is the effect of aerosols in the ozone DIAL measurements ? What are the

possible biases that they might induce ? Please provide an estimation of this bias for the cases of very high aerosol concentrations shown in Fig 3 on 22 March 2015.

11) Line 170 : What is the scientific evidence that suggest that small particles in the event of 22 March 2015 are "sulphate" particles ? This should be clearly justified.

12) Lines 174-181: Surface measurement of PM2.5 should be presented in the figure. A literal description seems insufficient. Please, add the time series of theses measurements.

13) Section 4.1: the model should be use to identify the type and origin of the aerosols.

14) Line 208: the term "AODs were almost the same" is only approximate. Objective terms should be used. Mean bias and RMS differences should also be given.

15) Trajectory analysis: Timing in the comparisons does not seem to be precise enough. The ozone and aerosol plume arriving to Saga lasts for a few hours only. This cannot be justified with high ozone and aerosol concentration during a large period of time (e.g. March 2015). These airmasses arriving to Saga on 22 March were likely located near the Beijing area on which date precisely? On that date, where ozone and aerosol concentrations high?

16) Lines 237-238: The justification of this statement "Based on these lidar data and the in-situ measurement data at Takagimachi in Saga city, an air mass with high ozone and aerosol concentrations could have been transported from the free troposphere" is not clear to me. The high concentrations are observed in the lidar time series below 1.5 km. There is not indication that transport of pollutants only occurs in the free troposphere, given that the top of the boundary layer is usually near 1.5 km. What evidence is given that transport does not occur in the boundary layer or both in the free troposphere and the boundary layer? What is the height of the mixing boundary layer (during, the day, the night, in the region). The altitude of the back-trajectories should also be given.

---

## Referee Comment (RC2) · Anonymous Referee #2 · 5 Aug 2016

Review of ACPD manuscript

The manuscript submitted to ACPD presents continuous lidar measurements of ozone and aerosol extinction at 532 and 1064 nm during 11 days over Southern Japan. The data are compared with two Japanese global models: MRI-CCM2 for ozone and MASINGAR-mk2 for aerosol. The analysis of this long lidar data set is worth publication. However the goals of the paper are not well established and as already noticed by reviewer 1 the interpretation does not provide enough details to be very useful. There are already many publications reporting high ozone episodes measured with lidar techniques and a new paper on this topic has to go beyond existing literature. I see two ways: either to make a complete use of the long record and the vertical information

available in the lidar data (e.g. comparing the 21-23 March with the 30-31 March ozone episode) or a true validation exercise of the models. In the present version neither option is really developed.

As suggested by reviewer 1, a focus on the vertical transport during the high ozone episode using planetary boundary layer (PBL) development from the Mie lidar and temporal evolution of the modelled plumes may be a good option. Comparison of the March 21-23 with the March 30-31 ozone episode could be also interesting because the aerosol properties look different and the AOD on March 30 is less than the very large AOD seen on March 22. If the model assessment is the preferred option, then proper metrics used in papers discussing model accuracy have to be applied (e.g. see AEROCOM web site). Statistical parameters to quantify the quality of the simulation (mean bias, RMSE, normalized mean bias, etc...) and a discussion of all the possible model error sources must be provided.

Therefore I propose a major update of the paper in either direction to avoid having just another report of a high ozone episode seen by a lidar.

Detailed comments

Introduction

A lot of details are given on GOSAT satellite validation which is not the scope of the paper.

Line 60 Provide references about previous work dealing with high ozone episode measured by lidar: Kuang et al. Atmos. Env. 2011, Banta et al. JGR 1998, Eisele and Trickl Appl.Optics 2005, Ancellet et al. Atmos.Res. 2005, Kourtidis et al. JGR 2002, . . .. I believe none of these papers have published an 11-day continuous record so the advantage of your data set could be better presented in the introduction.

Section 2

Technical details about the lidar system are already given in Uchino 2012 and 2014

so it can be shortened, and alternatively provide the measurements characteristics (vertical and temporal resolution, range for the different wavelengths, specific aerosol corrections, lowest measurement range ...). Are the ozone concentrations really given with 7.5 m and 1 min resolution ?

Section 3

line 93 Be more precise about the criteria used to remove data affected by aerosols or clouds. Do you apply any aerosol corrections before this quality check ? If yes describe this correction. If not how large will be the bias in the ozone retrieval with a AOD larger than 1 at 532 nm as seen on March 22 ?

line 94 Define Ox. I understand why Ox is useful to discuss photochemistry in NO2 rich environment, but why not reporting surface ozone in Figure 2 ?

line 103 The daily cycle which is clearly seen in surface Ox measurements is hardly visible in the lidar record even at the lower bound near 500 m. What is the reason for this ? Is the nocturnal PBL always lower than the lidar measurement range ?

Line 106. This statement is not supported by any analysis. At least references must be given to support such a statement and all the observational evidence must be provided to validate this hypothesis.

Line 126. What the uncertainty of the DIAL ozone data ? The discrepancy between the model O3 concentrations and the lidar data is very large (> 100%) which is generally well simulated by model (mean bias usually less than 50%). This should be discuss with more details to attribute the bias either to uncertainties in the data or to model error. Could you quantify the uncertainty related to the emission inventory ?

Section 4

line 139 What is the reference altitude where molecular scattering can be assumed ? On March 22 aerosol layers are seen up to 6 km and the Fernald inversion may be biased. This point must be addressed in the paper.

Line 140 Lidar ratio may change from 70 sr for pollution aerosol to 30 sr for dust or marine aerosol. What is the reason for choosing 50 sr ? Of course it is likely to increase again the model underestimate of the extinction on March 22 if you apply 70 sr. What are the lidar ratio values assumed in the model simulation ?

Line 155 I assume the authors are using the 1064 backscatter ratio to calculate the wavelength exponent alp but the error is generally large for the inversion in the IR. What is the expected error on alp ? How does this exponent change for the different layers observed in the lidar record.

Line 169 and 171 Provide error bar on the depolarization ratio and wavelength exponent.

Line 169 Instead of looking at the variability of depolarization and wavelength exponent between the high aerosol event with the cleaner atmosphere before and after, it is probably more relevant to compare the different aerosol plumes between each other (e.g. March 22 with March 30)

Line 173 Provide references when interpreting the variability of Dp and Alp.

Section 4

Line 204 Is 225 m the lowest lidar measurement ? See my previous comments. It is important to be clear about this when discussing exchanges between the PBL and the layers aloft.

Line 214 There are many possible error sources in aerosol models. A better discussion is needed here including the existing literature about aerosol model errors and specific error quantification of MASINGAR-mk2 which have been already published. It seems that dust concentration is overestimated while pollution aerosol concentration is underestimated when looking at the March 22 and 30 extinction..

Section 5

Line 240-241 I fully agree with this statement but it is not really developped in the paper.

---

## Author Comment (AC1) · 8 Nov 2016

The authors wish to thank two referees for helpful and thoughtful comments. Each comment is addressed individually below. The reference comments are written in black, and our responses are described in red.

The main changes of the paper since the original APCD versions are as follows:

- Figures were changed as follows: 1) Figs 3c, 4, and 7-11 were added, 2) Figs. 4 and 5 were replaced with Figs. 5 and 6, and 3) Fig. 6 was removed. In addition to these changes, the scale of Fig 3b was changed because total depolarization ratio was calculated only from the signal of the low sensitive P channel of which error was small.
- Fifteen references were added.
- Section 5 (Discussion and concluding remarks) was divided into two sections:
  5 Discussion: origin and transport pathways of ozone and aerosol plumes
  6 Concluding remarks

In the revised paper, the sentences moved from the original paper are written in blue. The original sentences in yellow highlighter with strikethrough were deleted in the revised paper. The original sentences with strikethrough were also deleted. The sentences added in the revised paper were written in red.

**Anonymous Referee #1**

The paper by Uchino et al. presents lidar measurements of high ozone and aerosol concentrations over southern Japan and a comparison with models. Measurements and the model outputs are interesting and a complete analysis of these datasets is worth scientific publication. However, the current analysis of these datasets does not seem to fully exploit the information they provide and many statements are not clearly justified (even though the observational and modelling datasets to do it are potentially available already). I recommend the following major revisions:

1) Evidence on the origins (location and type of source) and transport pathways (particularly in the vertical) of the ozone and aerosol plumes are not clearly provided. Chemistry-transport model outputs are not exploited for this, as currently, simulations are only used for comparison with lidars. I recommend fully developing this aspect, with analysis of the model outputs at the scale of East Asia and in the vertical. This is very important to provide information on the horizontal and vertical pathways of the pollutant plumes and their origin (e.g. type of aerosols).

We provided information on the horizontal and vertical pathways of the pollutant plumes and their origin including the type of aerosols in the discussion in lines 345-356 and 352-365, and revised the abstract in lines 24-33as follows:

Lines 347-356:

Figure 7 shows the time-altitude cross sections of total aerosol extinction coefficients at 550 nm, and the ratios of dust extinction coefficients to total aerosol extinction coefficients simulated by MASINGAR-mk2 with potential temperatures over Saga for 20–31 March 2015. For the event on 22 March, the model predicted the dust particles (60–100%) in the altitude range 1–3 km, and sulfate (40–60%) and dust (30–40%) particles below 1 km. The number of the parenthesis represents the ratio of each component's extinction coefficient to the total extinction coefficient. The dust particles descended to the surface in the afternoon (Fig.7b). For the event on 30 March, MASINGAR mk-2 predicted the dust particles (50–100%) for 1–6 km, and sulfate (50–80%) and dust (0–20%) particles below 1 km in the morning. Mie lidar data support the model prediction because $D_p$ is high (17 ± 6%) for 1–3 km and low (10 ± 3%) below 1 km. For both events, small amounts of organic carbon, black carbon and sea salt particles were predicted.

Lines 360-373:

An air parcels was  initially left at altitudes of  1.500 m (Fig. 8a) and 500 m (Fig.9a) over the lidar site at Saga. The trajectories were calculated for three days from 21:00 UTC  on 21 (06:00 JST on 22) March 2015. Figures 8b and 9b show the time-altitude cross sections of dust and sulfate extinction coefficients simulated by MASINGAR mk-2 along the trajectory paths of Figs. 8a and 9a respectively. Based on the results of the backward trajectories and the model simulations, the dust and sulfate particles  on 22  March could have been transported within about two days from the Gobi Desert and the North China Plain (NCP), respectively. The highest concentrations of $SO_2$ and $NO_2$ in the world were observed in NCP for 2013–2015 by the Ozone Monitoring Instrument (OMI) onboard NASA's Aura satellite, as shown in Fig.5 by Krotkov et al. (2016). These gases are important precursors of sulfate particles and ozone. Figure 10 represents the horizontal maps of ozone volume mixing ratios at 925 hPa (about 760 m altitude) simulated by MRI-CCM2 at 21:00 JST on 19, 20, 21 March and at 03:00 JST on 22 March 2015. These maps indicate that the high ozone could be transported from NCP to the Yellow Sea and then Saga within about two days.

Abstract, lines 24-33:

Backward trajectory analysis and the simulations by the Model of Aerosol Species IN the Global AtmospheRe (MASINGAR) mk-2 and the Meteorological Research Institute Chemistry-Climate Model, version 2 (MRI-CCM2) indicated that the mineral dust particles originated from the Gobi Desert and an air mass with high ozone and aerosol (mainly sulfate) concentrations

originated from the North China Plain  could have been transported over the measurement site within about two days.  These  high ozone and aerosol  concentrations impacted surface air quality substantially in the afternoon on 22 March 2015.

In accordance with these modifications, we added four figures (Figs.7–11) using the model outputs of HYSPLIT, MASINGAR-mk2 and MRI-CCM2, and lidar data.

2) Analysis of lidar data: the vertical structure of ozone and aerosol layers are not completely analysed, as the heights of the atmospheric boundary layer (mixing and residual layers) are not depicted. As shown in numerous papers, the detection of these layers may be easily done with the aerosol lidar. Such analysis should be added in the paper and precisely explain in a chronological order the mechanisms involved (i.e.vertical mixing, arrival of pollutants, etc).

We estimated the tops of the atmospheric boundary layers from the range-corrected backscatter signal at 1064 nm (Fig.11), and explained the arrival of pollutants (dust, sulfate and ozone) and their vertical mixing to the surface when the mixed layer developed in the afternoon in lines 383-400:

   To investigate the vertical transport processes of the aerosol and ozone in the lower troposphere over the measurement site, we show in Figure 11 the time variations of the top altitudes of the atmospheric boundary layers from 11:10 JST on 20 March to 14:33 JST on 31 March 2015 which were estimated from the 1064 nm range-corrected backscatter signals with a range resolution of 15 m using the wavelet covariance transform method (Baars et al., 2008; Izumi et al., 2016), and those obtained from the radiosonde data at Fukuoka and the JMA Meso-scale Analysis (MA) data over Saga using the parcel method (Holzworth, 1964). When the mixed layers developed in the afternoon, the tops of the mixed layers (1.5–2 km) estimated by Mie lidar were almost consistent with those by MA. However, Mie lidar had a tendency to detect the residual layers in the night and morning time, e.g., 21–22 March although the radiosonde data at 9:00 JST on 22 March found the top of the mixed layer was 117 m (Stull, 1988), and it was difficult for Mie lidar to detect the mixed layer because the lowest altitude of the Mie lidar measurement was 225 m.

   The dust particles originated from the Gobi Desert arrived at 1–3 km altitudes above the residual layer over the lidar site at 06:00 JST on 22 March. When the mixed layer developed to 1.5–2 km at 11:00–15:00 JST on 22, the dust particles were supposed to be mixed into the boundary layer and then reached the surface by the entrainment, as simulated in Fig.7b. This could result in the sharp increase in PM2.5 concentrations at the surface increased sharply after 11:00 JST, as shown in Fig.4. The similar phenomenon was observed over the northern Kyushu area during the dust event in late May–early June 2014 (Uno et al., 2016).

3) Technical characteristic of the datasets: in the presentation of the results, too much technical information of the datasets (instrument characteristics, model configuration, variables describing the

datasets, etc) is given. The geophysical interpretation of the measurements is relatively scarce. For sake of clearness for the readers, I recommend that most of technical remarks on the datasets are given a previous section dedicated to datasets characteristics and then only the geophysical interpretation is given when describing the figures.

We moved the remarks on the observation (formerly, physical) parameters obtained by Mie lidar and the vertical and time resolutions of Mie lidar and ozone DIAL data to Section 2.

4) The extinction-to-backscatter ratio : the climatological values that are used are expected to be suited to a particular type of aerosol (it changes a lot depending on the origin and size). What kind of aerosol is it for 50 sr at 532 nm? Measurements suggest the presence of large non-spherical particles (likely desert dust). How is this taken into account? I recommend using different extinction-to-backscatter ratios for each kind of particles (dust, sulphate, etc).

We corrected as follows:

Lnes 97-99:

We assumed the lidar ratio *LR* (extinction-to-backscatter ratio) for aerosols to be 50 sr at 532 nm and 45 sr at 1064 nm based on the lidar ratios for Asian dust and pollution aerosols summarized by Sakai et al. (2003), Anderson et al. (2003) and Cattrall et al. (2005).

Their summaries are as follows:
  Sakai et al.(2003): Asian dust 47±18 sr,
  Cattrall et al.(2005): Dust (spheroids) 42±4 sr, SE AsiaPollution 58±10 sr,
  Anderson et al.(2003): ACE-Asia Pollution
  (Fine-dominated, submicron portion) 50±5 sr, Dust (Coarse-dominated, Dust-like chemistry, Supermicron portion) 46±8 sr.

I recommend the following minor revisions:

1)   English language should be revised in the whole the paper.

Before we submitted our manuscript to ACP, the manuscript had been edited carefully by two native-English-speaking professional editors from ELSS, Inc. (elss@elss.co.jp, http://www.elss.co.jp). In the revised paper, we made our efforts for the readers to understand it more clearly.

2) Colour scales in figure 2 should be changed in order to highlight changes from background.

We highlighted the high ozone event on 22 March 2015 in red color.

3) Line 93: what is the meaning of "The errors were equated to 100% times the lidar signal-to-noise ratios". Please clarify.

We changed the sentence as follows:

Line 151-152:

The errors were computed from the lidar signal-to-noise ratios by use of Poisson statistics.

4) Lines 106-108 : The justification of the following statement is not clear : "These observational results show the descent of regions of high ozone concentrations and suggest that air with high ozone concentrations was transported to the surface by vertical mixing when the planetary boundary layer developed during the daytime". What is the height of the boundary layer? This height should be clearly shown in the figures. When was it mixed? Which elements suggest a "descent" or downward mixing of the high ozone plume? This should be thoroughly explained and justify. The model may be used to describe this mixing mechanism. When it is mentioned "daytime" vertical mixing, it is in which day? In which location vertical occurs? The current explanation is very scarce and unclear. This should be thoroughly explained and justify.

We deleted the sentence of "These observational results show the descent of regions of high ozone concentrations and suggest that air with high ozone concentrations was transported to the surface by vertical mixing when the planetary boundary layer developed during the daytime" in Section 3. Instead, we discussed on this matter in Section 5 using Figs. 11.

Lines 383-400:

To investigate the vertical transport processes of the aerosol and ozone in the lower troposphere over the measurement site, we show in Figure 11 the time variations of the top altitudes of the atmospheric boundary layers from 11:10 JST on 20 March to 14:33 JST on 31 March 2015 which were estimated from the 1064 nm range-corrected backscatter signals with a range resolution of 15 m using the wavelet covariance transform method (Baars et al., 2008; Izumi et al., 2016), and those obtained from the radiosonde data at Fukuoka and the JMA Meso-scale Analysis (MA) data over Saga using the parcel method (Holzworth, 1964). When the mixed layers developed in the afternoon, the tops of the mixed layers (1.5–2 km) estimated by Mie lidar were almost consistent with those by MA. However, Mie lidar had a tendency to detect the residual layers in the night and morning time, e.g., 21–22 March although the radiosonde data at 9:00 JST on 22 March found the top of the mixed layer was 117 m (Stull, 1988), and it was difficult for Mie lidar to detect the mixed layer because the lowest altitude of the Mie lidar measurement was 225 m.

The dust particles originated from the Gobi Desert arrived at 1–3 km altitudes above the residual layer over the lidar site at 06:00 JST on 22 March. When the mixed layer developed to 1.5–2 km at 11:00–15:00 JST on 22, the dust particles were supposed to be mixed into the boundary layer and then reached the surface by the entrainment, as simulated in Fig.7b. This could result in the sharp increase in PM2.5 concentrations at the surface increased sharply after 11:00 JST, as shown in Fig.4. The similar phenomenon was observed over the northern Kyushu area during the dust event in late May–early June 2014 (Uno et al., 2016).

5) Line 120: The sentence "The most reasonable results were obtained when the following changes were made (Fig. 2b)" does not seem to match the Figure as it does not explain the changes in the model, but it

presents the final results.

We changed the sentence in lines 181-183:

However, the MRI-CCM2 predicted the high ozone concentrations of 50–60 ppbv and could not reproduce those the high ozone concentrations of 90–110 ppbv observed below an altitude of 1.5 km during 03:00–20:00 JST on 22 March 2015.

6) Line 130: there is also a difference in the time of the arrival of the plume. Please, report.

We added the following sentence to lines 194-195:

And the MRI-CCM2 predicted the high ozone concentration a half day earlier than the DIAL observation.

7) The term "bad weather" is not objective. I suggest to use "rainy" conditions or similar.

We used "rainy and cloudy conditions".

8) The wavelength exponent, Alp : the name to this variable does not seem very conventional. For optical depth, the term "Ångström exponent" is usually used in literature. The name ´n wavelength exponent Â˙z does not seem very explicit. Why the symbol is "Alp" ? Is this term often used by the scientific community?

As suggested by the referee, the term " Ångström exponent" is usually used for optical depth in literature. Therefore we use "The backscatter-related Ångström exponent $Alp$" for backscattering coefficient (line 115-116).

9) Line 167 : The term "always existed" is not clear. Does it mean that it is observed in the timeseries shown in Figure 3 ? Please, clarify.

We added "for 20–31 March 2015" (lines 251-252).

10) What is the effect of aerosols in the ozone DIAL measurements ? What are the possible biases that they might induce ? Please provide an estimation of this bias for the cases of very high aerosol concentrations shown in Fig 3 on 22 March 2015.

We added the following sentences in lines 196-204:

The maximum bias (systematic error) of ozone DIAL data caused by aerosols was estimated to be 20% (15 ppbv) at 0.57 km, and the mean bias and the standard deviation were 7% ± 5% in the altitude range 0.57–2.0 km at 11:00 JST. These biases were estimated from $Alp$ observed at the same time by Mie lidar and assuming $LR$ =50 sr in the wavelength range 276–299 nm, based on the equations of (6) and (7) in Uchino and Tabata (1991). These biases were not large since the 276/287 nm and 287/299 nm wavelength pairs were suitable for measurements of ozone in the boundary layer and the free troposphere respectively (Nakazato et al., 2007). As mentioned earlier, the ozone DIAL data with the statistical error smaller than 10% was used in this study. Therefore the uncertainty of the ozone DIAL data was estimated to be smaller than 22% and the mean value of the uncertainty was 12%.

11) Line 170 : What is the scientific evidence that suggest that small particles in the event of 22 March 2015 are "sulphate" particles ? This should be clearly justified.

Because we did not make the in-situ measurement of aerosol component, we deleted the word "sulfate" from the text (line 257). However, the model predicted that the particles were mostly sulfate.

12) Lines 174-181: Surface measurement of PM2.5 should be presented in the figure. A literal description seems insufficient. Please, add the time series of theses measurements.

We added "Fig.4" to show hourly data of surface PM2.5 and Ox at Takagimachi, Saga for 20-31 March 2015.

13) Section 4.1: the model should be use to identify the type and origin of the aerosols.

We used the model to identify the type and origin of the aerosols. The results were presented in Figs. 7, 8 and 9 and the explanation was provided lines 347-367:

Figure 7 shows the time-altitude cross sections of total aerosol extinction coefficients at 550 nm, and the ratios of dust extinction coefficients to total aerosol extinction coefficients simulated by MASINGAR-mk2 with potential temperatures over Saga for 20–31 March 2015. For the event on 22 March, the model predicted the dust particles (60–100%) in the altitude range 1–3 km, and sulfate (40–60%) and dust (30–40%) particles below 1 km. The number of the parenthesis represents the ratio of each component's extinction coefficient to the total extinction coefficient. The dust particles descended to the surface in the afternoon (Fig.7b). For the event on 30 March, MASINGAR mk-2 predicted the dust particles (50–100%) for 1–6 km, and sulfate (50–80%) and dust (0–20%) particles below 1 km in the morning. Mie lidar data support the model prediction because $D_p$ is high (17 ± 6%) for 1–3 km and low (10 ± 3%) below 1 km. For both events, small amounts of organic carbon, black carbon and sea salt particles were predicted.

To identify the origin of the  aerosols and related transport processes, three-dimensional backward trajectories of air parcels were calculated with the NOAA Hybrid Single Particle Lagrangian Integrated Trajectory (HYSPRIT) model (Draxler and Hess, 1998; Stein et al., 2015). An air parcel was  initially left at altitudes of  1500 m (Fig. 8a) and 500 m (Fig.9a) over the lidar site at Saga. The trajectories were calculated for three days from 21:00 UTC  on 21 (06:00 JST on 22) March 2015. Figures 8b and 9b show the time-altitude cross sections of dust and sulfate extinction coefficients simulated by MASINGAR mk-2 along the trajectory paths of Figs. 8a and 9a respectively. Based on the results of the backward trajectories and the model simulations, the dust and sulfate particles  on 22  March could have been transported within about two days from the Gobi Desert and the North China Plain (NCP), respectively, to the measurement site .

14) Line 208: the term "AODs were almost the same" is only approximate. Objective

terms should be used. Mean bias and RMS differences should also be given.

We added the following sentence in lines 307-309:

The mean bias and the standard deviation of AOD between Mie lidar and sky radiometer was 0.029 ± 0.051, and that between MASINGAR mk-2 and sky radiometer was –0.07 ± 0.24 for 20–31 March, except for 12:00–14:00 on 22 March.

15) Trajectory analysis: Timing in the comparisons does not seem to be precise enough. The ozone and aerosol plume arriving to Saga lasts for a few hours only. This cannot be justified with high ozone and aerosol concentration during a large period of time (e.g. March 2015). These airmasses arriving to Saga on 22 March were likely located near the Beijing area on which date precisely? On that date, where ozone and aerosol concentrations high?

We added the following sentence in lines 374-375:

Because it was difficult to obtain observational data of surface ozone and sulfate particles in NCP including Beijing on 19-20 March, we refer to the following papers related to those data.

16) Lines 237-238: The justification of this statement "Based on these lidar data and the in-situ measurement data at Takagimachi in Saga city, an air mass with high ozone and aerosol concentrations could have been transported from the free troposphere" is not clear to me. The high concentrations are observed in the lidar time series below 1.5 km. There is not indication that transport of pollutants only occurs in the free troposphere, given that the top of the boundary layer is usually near 1.5 km. What evidence is given that transport does not occur in the boundary layer or both in the free troposphere and the boundary layer? What is the height of the mixing boundary layer (during, the day, the night, in the region). The altitude of the back-trajectories should also be given.

We estimated the tops of the atmospheric boundary layers from the range-corrected backscatter signal at 1064 nm (Fig.11), and explained the arrival of pollutants (dust, sulfate and ozone) and their vertical mixing to the surface when the mixed layer developed in the afternoon in lines 383-404 (The texts were given in the response to the referee's comment 4).
The manuscript submitted to ACPD presents continuous lidar measurements of ozone and aerosol extinction at 532 and 1064 nm during 11 days over Southern Japan. The data are compared with two

Japanese global models: MRI-CCM2 for ozone and MASINGAR-mk2 for aerosol. The analysis of this long lidar data set is worth publication. However the goals of the paper are not well established and as already noticed by reviewer 1 the interpretation does not provide enough details to be very useful. There are already many publications reporting high ozone episodes measured with lidar techniques and a new paper on this topic has to go beyond existing literature. I see two ways: either to make a complete use of the long record and the vertical information available in the lidar data (e.g. comparing the 21-23 March with the 30-31 March ozone episode) or a true validation exercise of the models. In the present version neither option is really developed. As suggested by reviewer 1, a focus on the vertical transport during the high ozone episode using planetary boundary layer (PBL) development from the Mie lidar and temporal evolution of the modelled plumes may be a good option. Comparison of the March 21-23 with the March 30-31 ozone episode could be also interesting because the aerosol properties look different and the AOD on March 30 is less than the very large AOD seen on March 22. If the model assessment is the preferred option, then proper metrics used in papers discussing model accuracy have to be applied (e.g. see AEROCOM web site). Statistical parameters to quantify the quality of the simulation (mean bias, RMSE, normalized mean bias, etc...) and a discussion of all the possible model error sources must be provided. Therefore I propose a major update of the paper in either direction to avoid having just another report of a high ozone episode seen by a lidar.

The authors wish to thank the referee for helpful and thoughtful comments. In accordance with the referee's comment, we present the comparison of the March 22 with the March 30 episode because the aerosol properties look different and the AOD on March 30 is less than the very large AOD seen on March 22. To describe this, we added the following sentences in lines 347-356:

Figure 7 shows the time-altitude cross sections of total aerosol extinction coefficients at 550 nm, and the ratios of dust extinction coefficients to total aerosol extinction coefficients simulated by MASINGAR-mk2 with potential temperatures over Saga for 20–31 March 2015. For the event on 22 March, the model predicted the dust particles (60–100%) in the altitude range 1–3 km, and sulfate (40–60%) and dust (30–40%) particles below 1 km. The number of the parenthesis represents the ratio of each component's extinction coefficient to the total extinction coefficient. The dust particles descended to the surface in the afternoon (Fig.7b). For the event on 30 March, MASINGAR mk-2 predicted the dust particles (50–100%) for 1–6 km, and sulfate (50–80%) and dust (0–20%) particles below 1 km in the morning. Mie lidar data support the model prediction because $D_p$ is high (17 ± 6%) for 1–3 km and low (10 ± 3%) below 1 km. For both events, small amounts of organic carbon, black carbon and sea salt particles were predicted.

Detailed comments

Introduction

A lot of details are given on GOSAT satellite validation which is not the scope of the paper. Line 60

Provide references about previous work dealing with high ozone episode measured by lidar: Kuang et al. Atmos. Env. 2011, Banta et al. JGR 1998, Eisele and Trickl Appl.Optics 2005, Ancellet et al. Atmos.Res.

2005, Kourtidis et al. JGR 2002, . . .. I believe none of these papers have published an 11-day continuous record so the advantage of your data set could be better presented in the introduction.

We added the following sentences in lines 69-72:

High ozone episodes in the lower troposphere have been observed by lidar (Banta et al., 1998; Koutidis et., 2002; Ancellet et al., 2005; Eisele and Trickl, 2005; Kuang et al., 2011). These observation records were limited to one week at most. We made an 11-day continuous record on 20–31 March 2015.

Section 2

Technical details about the lidar system are already given in Uchino 2012 and 2014 so it can be shortened, and alternatively provide the measurements characteristics (vertical and temporal resolution, range for the different wavelengths, specific aerosol corrections, lowest measurement range . . .). Are the ozone concentrations really given with 7.5 m and 1 min resolution ?

As suggested, we revised the texts in lines 135-139 as follows:

The 276/287 nm and 287/299 nm wavelength pairs were used for ozone DIAL measurements in the altitude ranges of 0.57–2.0 km and 2.0–6.0 km, respectively. The effective vertical resolutions were 270 m for 0.57–2.0 km and 540 m for 2.0–6.0 km, respectively (Uchino et al., 2014). The time resolution was set to 1 h to facilitate comparison with the MRI-CCM2. The aerosol correction was not made for the ozone retrieval.

Section 3

line 93 Be more precise about the criteria used to remove data affected by aerosols or clouds. Do you apply any aerosol corrections before this quality check ? If yes describe this correction. If not how large will be the bias in the ozone retrieval with a AOD larger than 1 at 532 nm as seen on March 22 ?

The criterion was added in lines 152-154:

Regions surrounded by a black rectangle are areas where the data were affected by aerosols and/or clouds with $R$ larger than 2 at 299 nm, which were calculated assuming $LR$=50 sr without correcting attenuation by ozone absorption.

We did not apply any aerosol corrections, and so added the following sentence in lines 138-139:
The aerosol correction was not made for the ozone retrieval.

For the bias of ozone DIAL data, we added the following sentences in 196-204:

The maximum bias (systematic error) of ozone DIAL data caused by aerosols was estimated to be 20% (15 ppbv) at 0.57 km, and the mean bias and the standard deviation were 7% ± 5% in the altitude range 0.57–2.0 km at 11:00 JST. These biases were estimated from $Alp$ observed at the same time by Mie lidar and assuming $LR$ =50 sr in the wavelength range 276–299 nm, based on the equations of (6) and (7) in Uchino and Tabata (1991). These biases were not large since the 276/287 nm and 287/299 nm wavelength pairs were suitable for measurements of ozone in the boundary layer and the free troposphere respectively

(Nakazato et al., 2007). As mentioned earlier, the ozone DIAL data with the statistical error smaller than 10% was used in this study. Therefore the uncertainty of the ozone DIAL data was estimated to be smaller than 22% and the mean value of the uncertainty was 12%.

line 94 Define Ox. I understand why Ox is useful to discuss photochemistry in NO2 rich environment, but why not reporting surface ozone in Figure 2 ?

We mentioned as follows in lines 158-160:

Because the contribution of other components such as peroxyacetyle nitrate (PAN) to oxidant concentrations was extremely low, the oxidant volume mixing ratio was considered to be that of ozone.

line 103 The daily cycle which is clearly seen in surface Ox measurements is hardly visible in the lidar record even at the lower bound near 500 m. What is the reason for this ? Is the nocturnal PBL always lower than the lidar measurement range ?

We detected the low ozone concentrations in the nighttime on 10–11 January 2013, but the daily cycle of ozone was not detected at the lower bound near 600 m in this DIAL observation record. The daily cycle of ozone predicted by MRI-CCM2 was clear at least up to 250 m but not clear at 600 m. Therefore we are interested in the DIAL system which can measure ozone in the altitude range 100–500 m.

Line 106. This statement is not supported by any analysis. At least references must be given to support such a statement and all the observational evidence must be provided to validate this hypothesis.

In accordance with the reviewer's comment, the statement was deleted from Section 3. Instead we presented the related lidar observation results, the model output of MASINGAR-mk2, and the references in Discussion.

Line 126. What the uncertainty of the DIAL ozone data ? The discrepancy between the model O3 concentrations and the lidar data is very large (> 100%) which is generally well simulated by model (mean bias usually less than 50%). This should be discuss with more details to attribute the bias either to uncertainties in the data or to model error. Could you quantify the uncertainty related to the emission inventory ?

For the bias of ozone DIAL data, we added the following sentences in 196-204:

The maximum bias (systematic error) of ozone DIAL data caused by aerosols was estimated to be 20% (15 ppbv) at 0.57 km, and the mean bias and the standard deviation were 7% $\pm$ 5% in the altitude range 0.57–2.0 km at 11:00 JST.    These biases were estimated from *Alp* observed at the same time by Mie lidar and assuming *LR* =50 sr in the wavelength range 276–299 nm, based on the equations of (6) and (7) in Uchino and Tabata (1991). These biases were not large since the 276/287 nm and 287/299 nm wavelength pairs were suitable for measurements of ozone in the boundary layer and the free troposphere respectively (Nakazato et al., 2007). As mentioned earlier, the ozone DIAL data with the statistical error smaller than 10% was used in this study. Therefore the uncertainty of the ozone DIAL data was estimated to be smaller

than 22% and the mean value of the uncertainty was 12%.

For the uncertainty of the inventory of sulfur dioxide ($SO_2$), we added the following sentences in lines 320-327:
Grainer et al.(2011) collected various emission inventories and compared them in global scales. They found that differences in Chinese sulfur dioxide ($SO_2$) emissions in 2000 reached 66% between the lowest and highest emissions and concluded that there was no consensus among the different inventories for the emissions of Chinese $SO_2$. This large variation among the inventories indicates that estimate of $SO_2$ emission in China has large error. In their comparison, the MACCity emission which was used in MASINGAR-mk2 simulation, showed the lower amount of Chinese $SO_2$ emission among the inventories. This fact might be responsible for the underestimation of pollution aerosol (sulfate) concentrations.

Section 4

line 139 What is the reference altitude where molecular scattering can be assumed ? On March 22 aerosol layers are seen up to 6 km and the Fernald inversion may be biased. This point must be addressed in the paper.
We added the following sentences in lines 311-315:
One possible reason for the large difference in AOD (~0.2) between Mie lidar and Sky radiometer data is that we set the reference altitudes 8.2 km and 2.8 km at 12:00 and 13:00 JST on 22 March, respectively, for the lidar because the backscattered signals were strongly attenuated by the dense aerosol layers below 2 km. This might cause the large differences of AODs between Mie lidar and sky radiometer data.

Line 140 Lidar ratio may change from 70 sr for pollution aerosol to 30 sr for dust or marine aerosol. What is the reason for choosing 50 sr ? Of course it is likely to increase again the model underestimate of the extinction on March 22 if you apply 70 sr. What are the lidar ratio values assumed in the model simulation ?
The lidar ratio *LR* is not used in the MASINGAR-mk2 because the extinction coefficient is calculated directly from the size distribution and the refractive index of each aerosol component.

Line 155 I assume the authors are using the 1064 backscatter ratio to calculate the wavelength exponent alp but the error is generally large for the inversion in the IR. What is the expected error on alp ? How does this exponent change for the different layers observed in the lidar record.
We added the following sentences in lines 260-264:
When there were no clouds above, *R* at 1064 nm was estimated assuming *Alp*=1.5 at the reference altitude where very small amount of aerosols was expected to be present, i.e., R=1.06 ± 0.06 (D=1.2 ± 0.51) at 532 nm, in the altitude range 3–6 km. If the value of *Alp* was changed from 1.0 to 2.0 at the reference altitude, the uncertainty in *Alp* was estimated to be ± 0.2. Alp was 0.3–2.0 in the 11-day Mie lidar record.

Line 169 and 171 Provide error bar on the depolarization ratio and wavelength exponent.

We added the error bar (one standard deviation) on the depolarization ratio and wavelength exponent in lines 252-265:

An event of high aerosol loading with large values of $R$ ($>8$ ) was observed below altitudes of 1.5 km for 03:00–21:00 JST on 22 March, when the values of $D$ were small (the mean and the standard deviation: $3.9 \pm 2.1$ %) compared with those before and after the event, when the values of $D$ were larger than $7.9 \pm 2.1$ % during 15:00 JST on 21 through 15:00 JST on 23, except 03:00–21:00 JST on 22. The main aerosol component during the event might be  submicrometer-sized spherical  particles, because $D_p$ was small ($4 \pm 2$ %), and the wavelength exponent $Alp$ was large ($1.3 \pm 0.3$ ). In contrast, the main aerosol particles before and after the event could  be supermicrometer-sized, nonspherical mineral dust particles because $D_p$ was comparatively large ($13 \pm 3$ %) and $Alp$ was  $\pm 0.2$ (Sakai et al., 2003; Cattrall et al., 2005) . When there were no clouds above, $R$ at 1064 nm was estimated assuming $Alp=1.5$ at the reference altitude where very small amount of aerosols was expected to be present, i.e., $R=1.06 \pm 0.06$ ($D=1.2 \pm 0.51$) at 532 nm, in the altitude range 3–6 km. When the value of $Alp$ was changed from 1.0 to 2.0 at the reference altitude, the uncertainty in $Alp$ was estimated to be $\pm 0.2$. The maximum errors of $D$ and $D_p$ were 0.1% and 2% for $R>2$ at 532 nm.

Line 169 Instead of looking at the variability of depolarization and wavelength exponent between the high aerosol event with the cleaner atmosphere before and after, it is probably more relevant to compare the different aerosol plumes between each other (e.g. March 22 with March 30)

We denoted the variability of Dp and Alp for the two events as follows:

Lines 257-258:

"$D_p$ was small ($4 \pm 2$ %), and the wavelength exponent $Alp$ was large ($1.3 \pm 0.3$ )" for altitudes of 1.5 km for 03:00–21:00 JST on 22 March.

Line: 355:

"$D_p$ is high ($17 \pm 6$%) for 1–3 km" on 30 March

Line 173 Provide references when interpreting the variability of Dp and Alp.

We provided two references of Sakai et al. (2003) and Cattrall et al. (2005) in line 258.

Section 4

Line 204 Is 225 m the lowest lidar measurement ? See my previous comments. It is important to be clear about this when discussing exchanges between the PBL and the layers aloft.

We added the following sentence in lines 124-125.

The lowest altitude of Mie lidar measurement was 225 m due to the non-perfect overlap of the transmitter-receiver optical axes of the lidar system.

Line 214 There are many possible error sources in aerosol models. A better discussion is needed here including the existing literature about aerosol model errors and specific error quantification of MASINGAR-mk2 which have been already published. It seems that dust concentration is overestimated while pollution aerosol concentration is underestimated when looking at the March 22 and 30 extinction.

We added the following sentences in lines 319-332

The other plausible reason for the underestimation is the uncertainty of the  emissions inventories of aerosol precursors. Grainer et al.(2011) collected various emission inventories and compared them in global scales. They found that differences in Chinese sulfur dioxide ($SO_2$) emissions in 2000 reached 66% between the lowest and highest emissions and concluded that there was no consensus among the different inventories for the emissions of Chinese $SO_2$. This large variation among the inventories indicates that estimate of $SO_2$ emission in China has large error. In their comparison, the MACCity emission which was used in MASINGAR-mk2 simulation, showed the lower amount of Chinese $SO_2$ emission among the inventories. This fact might be responsible for the underestimation of pollution aerosol (sulfate) concentrations. In MASINGAR-mk2, dust emission flux is estimated by a parameterized dust emission scheme and has strong dependency upon various parameters (i.e., soil texture, soil wetness, land use, snow cover fraction, vegetation cover, surface wind speed, etc.). The dust model intercomparison project (DMIP; Uno et al., 2006) reported that simulated dust emission amounts over East Asia among eight dust models (including the former version of MASINGAR) differed sometimes by a factor of ten. These facts indicate that estimate of dust emission also causes large errors.

Section 5
Line 240-241 I fully agree with this statement but it is not really developped in the paper.

We discussed on this matter in lines 383-400:

To investigate the vertical transport processes of the aerosol and ozone in the lower troposphere over the measurement site, we show in Figure 11 the time variations of the top altitudes of the atmospheric boundary layers from 11:10 JST on 20 March to 14:33 JST on 31 March 2015 which were estimated from the 1064 nm range-corrected backscatter signals with a range resolution of 15 m using the wavelet covariance transform method (Baars et al., 2008; Izumi et al., 2016), and those obtained from the radiosonde data at Fukuoka and the JMA Meso-scale Analysis (MA) data over Saga using the parcel method (Holzworth, 1964). When the mixed layers developed in the afternoon, the tops of the mixed layers (1.5–2 km) estimated by Mie lidar were almost consistent with those by MA. However, Mie lidar had a tendency to detect the residual layers in the night and morning time, e.g., 21–22 March although the radiosonde data at 9:00 JST on 22 March found the top of the mixed layer was 117 m (Stull, 1988), and it was difficult for Mie lidar to detect the stable boundary layer because the lowest altitude of the Mie lidar measurement was 225 m.

The dust particles originated from the Gobi Desert arrived at 1–3 km altitudes above the residual layer over the lidar site at 06:00 JST on 22 March. When the mixed layer developed to 1.5–2 km at 11:00–15:00 JST on 22, the dust particles were supposed to be mixed into the boundary layer and then

reached the surface by the entrainment, as simulated in Fig.7b. This could result in the sharp increase in PM2.5 concentrations at the surface increased sharply after 11:00 JST, as shown in Fig.4. The similar phenomenon was observed over the northern Kyushu area during the dust event in late May–early June 2014 (Uno et al., 2016).

lines 415-418:

Based on these lidar data,  the in-situ measurement data and the model simulation by MASINGAR -mk2 at Takagimachi in Saga city, there is a possibility that the air mass with high ozone and aerosol concentrations could have been transported from the lower troposphere to the surface by vertical mixing when the planetary boundary layer developed in the afternoon of 22 March 2015.

---

## Author Response (AR3)

[revised manuscript text omitted]

sr at 1064 nm based on the lidar ratios for Asian dust and pollution aerosols summarized by Sakai et al.

(2003), Anderson et al. (2003) and Cattrall et al. (2005). Their summaries are as follows: Sakai et al.

(2003): Asian dust 47 ± 18 sr, Cattral et al. (2005): dust (spheroids) 42 ± 4 sr, South East Asia pollution

[revised manuscript text omitted]
 cause error in AODs for the Mie lidar data. The sky radiometer could have different sight than the Mie lidar. This might be also a possible reason for the difference.

The model underestimated the AODs by factors of about 3.6–4 compared to the sky radiometer and lidar observations. One plausible reason for that is that the model resolution (about 60 km) was insufficient to reproduce the observed prominent peak in which the observed AOD increased from 1.0 to

2.0 in 6 hours. The other plausible reason for the underestimation is the uncertainty of the emissions inventories of aerosol precursors. Grainer et al. (2011) collected various emission inventories and compared them in global scales. They found that differences in Chinese sulfur dioxide ($SO_2$) emissions in

2000 reached 66% between the lowest and highest emissions and concluded that there was no consensus among the different inventories for the emissions of Chinese $SO_2$. This large variation among the inventories indicates that estimate of $SO_2$ emission in China has large error. In their comparison, the

MACCity emission which was used in MASINGAR-mk2 simulation, showed the lower amount of

Chinese $SO_2$ emission among the inventories. This might be responsible for the underestimation of pollution aerosol (sulfate) concentrations. In MASINGAR-mk2, dust emission flux is estimated by a parameterized dust emission scheme and has strong dependency upon various parameters (i.e., soil texture, soil wetness, land use, snow cover fraction, vegetation cover, surface wind speed, etc.). The dust model intercomparison project (DMIP; Uno et al., 2006) reported that simulated dust emission amounts over East Asia among eight dust models (including the former version of MASINGAR) differed sometimes by a factor of ten. These facts indicate that estimate of dust emission also causes 
[revised manuscript text omitted]

(a)

[Figure]

(b)

[Figure]

**Figure 9.** Same as Fig. 8 but for 500 m AGL.

(a)                                                    (b)

[Figure]

(c)                                                    (d)

**Figure 10.** Horizontal maps of ozone volume mixing ratios in ppbv predicted by MRI-CCM2 for 925 hPa
pressure level (about 760 m altitude) at 21:00 JST (JST=UT+9) on (a) 19, (b) 20, (c) 21 March and (d)
03:00 JST on 22 March 2015.

Old figure (delete)

[Figure]

.

Revised figure

[Figure]

**Figure 11.** Time-altitude cross section of (color shaded) range-corrected backscatter signal at 1064 nm and the tops of the mixed<s>atmospheric boundary</s> layers estimated by Mie lidar (closed black circles), radiosonde (open triangles), and JMA Meso-Scale Meteorological Analysis (open squares) data over Saga from 11:10 JST on 20 March to 14:33 JST on 31 March 2015.

Journal: ACP

Title: Lidar detection of high concentrations of ozone and aerosol transported from Northeast Asia over Saga, Japan

Author(s): Osamu Uchino et al.

MS No.: acp-2016-520

MS Type: Research article

Iteration: Minor Revision

**Reply to Co-Editor comments and Anonymous Referee #1 comments**

**Co-Editor Decision: Reconsider after minor revisions (Editor review)** (01 Dec 2016) by Yugo Kanaya

Comments to the Author:

Dear Authors,

Thank you for submitting your revised manuscript. Re-evaluation was made by our two reviewers. One suggested accepted as is, and the other requested minor revision. I would appreciate you could further respond to the comments by the reviewer #2. Also, I hope if you could take into account the handling editor's comments below:

The authors wish to thank the Co-Editor and the reviewer #2 for helpful and thoughtful comments. Each comment is addressed individually below. The reference comments are written in black, and our responses are described in red.

1.Although the focus of the manuscript is successfully shifted to the aspect of transport analysis, the Introduction part remained focused on the technical aspects and the authors' works. It is better to add some more general description on O3 and aerosols pollutions reaching this region in spring from the continent, citing literature (including those not using lidars), touching on what was studied before and what is yet to be analyzed (e.g., three dimensional features of ozone). This will highlight the value of this manuscript.

As suggested, we added the next sentences and 5 references:

Lines 45-50 and line 77 in the revised paper:

The aerosols transported from the East Asia to the western Japan were observed by lidar and their vertical distributions were reported (Iwasaka et al., 1988: Murayama et al., 2001; Hara et al., 2009). On the other hand the ozone pollutions from the Asia were mainly studied by the surface measurements (Akimoto et al., 1996; Yamaji et al., 2006). Continuous ozone vertical distributions by ozone DIAL are very useful for studying the transport process and the origin.

In this paper we report an event during which high concentrations of ozone and aerosols were observed almost simultaneously below an altitude of 1.5 km over Saga on 22 March 2015 by Mie lidar and ozone DIAL, which substantially impacted surface air quality. We also compared the observational results with those simulated by the models.

2. lines 150-151. Is an old-type ozone monitor based on wet chemistry is still under use at this site, where the PAN interference matters?

Ozone monitor is a UV photometer, and we revised as follows:

Lines 159-161:

Because the surface Ox was observed by an UV photometer, the contribution of other components such as peroxyacetyle nitrate (PAN) to oxidant concentrations was extremely low, and the oxidant volume mixing ratio was considered to be that of ozone.

3. line 181. Add base year of REAS2.1 emission inventory used for modeling.

We added "the REAS 2.1 emission inventory in 2007" in line 188.

4. lines 186 and 194. How is the 20% of systematic error related to the uncertainty of 12% in line 194?
We estimated as follows:
uncertainty$^2$=(systematic error)$^2$ + (statistical error)$^2$
the uncertainty: $(0.2^2+0.1^2)^{1/2}$=0.22
the mean value of the uncertainty: $(0.07^2+0.1^2)^{1/2}$=0.12

5. lines 266-267. The skyradiometer could have different sight than the lidar. This is also a possible reason for the difference.

We added "The sky radiometer could have different sight than the Mie lidar. This might be also a possible reason for the difference." in lines 276-277.

6. line 296. When the contribution from dust particles is 100%, those from others need to be 0%.

We replaced "(60–100%)", "(40–60%)", and "(30–40%)" in lines 306-307 with "(about 60–100%)", "(about 40–60%)", and "(about 30–40%)", respectively, and also corrected line 310 in the same manner.

7. Figure 8(b). There are several regions with increased dust extinction coefficient. Where do the authors think are the source regions of dust? The maximum during 1800 20 March - 0000 21 March is best connected to the increase over Saga; however, the region is over the ocean or the Shandong Peninsula and thus is not likely the source region.

Meteorological fields (e.g., wind velocities and pressure levels) used in the simulation of dust extinction coefficient (i.e., MASINGAR mk-2) are not identical to those used for the backward trajectory (i.e., HYSPLIT). This could explain that there are several regions where dust extinction coefficients are increased. Please find the next figure of dust emission simulated by MASINGAR mk-2 on 19 March (in UTC).

Based on this figure, we added the following sentences in lines 323-325:

The MASINGAR mk-2 simulation suggested that the dust particles emitted during 18:00–24:00 UTC on 19 March around 40°N, 105°E were responsible for the dust storm captured by the Mie lidar observation.

[Figure]

Figure S1, Daily averaged dust emission on 19 March. Red line shows the HYSPLIT-model backward trajectory from Saga at 1500 m above ground level (AGL) ending at 06:00JST on 22 May 2015 (identical to the backward trajectory shown in Figure 8).

**Submitted on 30 Nov 2016**
**Anonymous Referee #2**

In the revised version of the manuscript, a good effort has been done to improve the quality of the paper. Most of the corrections suggested were properly addressed. Before the acceptance of the paper, I recommend to address the following two minor points:

1)The altitudes detected from the lidar measurements (shown in Fig. 11) only in some cases correspond to the atmospheric boundary layer top. For numerous profiles (e.g. from 1200 JST of 20 March to 12 00 JST of 21 March), the method detects altitudes above 2 km that are not realistic boundary layer heights for this season. The algorithm applied to lidar measurements detects vertical gradients of aerosol backscattering, thus the top of aerosol layers that may not be the boundary layer top. A physical identification of the boundary layer top is done by recognizing the temperature inversion from radiosoundings as already shown in the figure. My recommendation is to screen out all heights detected from lidar data that do not correspond to the boundary layer height shown by the radiosoundings. Also, the difference between residual layer and mixing layer does not seem to be clear in the current paper. Radiosounding can tell the location of both as below the mixing layer the profile of potential temperature is neutral above an unstable layer near the surface and the residual layer corresponds to a neutral profile above a stable layer.

As suggested, Mie lidar had a tendency to detect not realistic mixed layer heights in the nighttime. Therefore we estimated the top altitudes of the mixed layers from two hours after sunrise to two hours before sunset, and revised in lines 342-344 as follows:

"we show in Fig. 11 the time variations of the top altitudes of the mixed layers from two hours after sunrise to two hours before sunset during 11:10 JST on 20 March through 14:33 JST on 31 March 2015", and we replaced the old Fig.11 with the revised Fig.11.

And we deleted "However, Mie lidar had a tendency to detect the residual layers in the night and morning time, e.g., 21–22 March." in lines 349-350 and " above the residual layer" in line 353.

2) The lidar ratio for both dust and pollution aerosols is the same (50 sr). This is an approximation as shown by the authors themselves: dust seem to have lower lidar ratios than pollution aerosol. I recommend to include in the text of the manuscript the values given in the answers to the reviewers : "Their summaries are as follows: Sakai et al.(2003): Asian dust $47\pm18$ sr, Cattrall et al.(2005): Dust (spheroids) $42\pm4$ sr, SE AsiaPollution $58\pm10$ sr, Anderson et al.(2003): ACE-Asia Pollution (Fine-dominated, submicron portion) $50\pm5$ sr, Dust (Coarse-dominated, Dust-like chemistry, Supermicron portion) $46\pm8$ sr." and then indicate that as a simplification, they use the same value for both species.

As suggested, the next sentences are added in lines 102-106:

Their summaries are as follows: Sakai et al. (2003): Asian dust $47 \pm 18$ sr, Cattral et al. (2005): dust (spheroids) $42 \pm 4$ sr, South East Asia pollution $58 \pm 10$ sr, Anderson et al. (2003): ACE-Asia pollution (fine-dominated, submicron portion) $50 \pm 5$ sr, dust (coarse-dominated, dust-like chemistry, supermicron portion) $46 \pm 8$ sr. As a simplification, we used the same value for both species.